# Photorealistic Text-to-Image Diffusion Models with Deep Language Understanding

**Chitwan Saharia**[*], **William Chan**[*], **Saurabh Saxena**[†], **Lala Li**[†], **Jay Whang**[†],
**Emily Denton, Seyed Kamyar Seyed Ghasemipour, Burcu Karagol Ayan,**
**S. Sara Mahdavi, Raphael Gontijo-Lopes, Tim Salimans,**
**Jonathan Ho**[†]**, David J Fleet**[†‡]**, Mohammad Norouzi**[*]

{sahariac,williamchan,mnorouzi}@google.com
{srbs,lala,jwhang,jonathanho,davidfleet}@google.com

Google Research, Brain Team
Toronto, Ontario, Canada

## Abstract

We present Imagen, a text-to-image diffusion model with an unprecedented degree of photorealism and a deep level of language understanding. Imagen builds on the power of large transformer language models in understanding text and hinges on the strength of diffusion models in high-fidelity image generation. Our key discovery is that generic large language models (e.g. T5), pretrained on text-only corpora, are surprisingly effective at encoding text for image synthesis: increasing the size of the language model in Imagen boosts both sample fidelity and image-text alignment much more than increasing the size of the image diffusion model. Imagen achieves a new state-of-the-art FID score of 7.27 on the COCO dataset, without ever training on COCO, and human raters find Imagen samples to be on par with the COCO data itself in image-text alignment. To assess text-to-image models in greater depth, we introduce DrawBench, a comprehensive and challenging benchmark for text-to-image models. With DrawBench, we compare Imagen with recent methods including VQ-GAN+CLIP, Latent Diffusion Models, GLIDE and DALL-E 2, and find that human raters prefer Imagen over other models in side-by-side comparisons, both in terms of sample quality and image-text alignment.

## 1 Introduction

Multimodal learning has come into prominence recently, with text-to-image synthesis [55, 12, 59] and image-text contrastive learning [51, 32, 77] at the forefront. These models have transformed the research community and captured widespread public attention with creative image generation [22, 56] and editing applications [21, 43, 36]. To pursue this research direction further, we introduce Imagen, a text-to-image diffusion model that combines the power of transformer language models (LMs) [15, 54] with high-fidelity diffusion models [28, 29, 16, 43] to deliver an unprecedented degree of photorealism and a deep level of language understanding in text-to-image synthesis. In contrast to prior work that uses only image-text data for model training [e.g., 55, 43], the key finding behind Imagen is that text embeddings from large LMs [54, 15], pretrained on text-only corpora, are remarkably effective for text-to-image synthesis. See Fig. 1 for select samples.

Imagen comprises a frozen T5-XXL [54] encoder to map input text into a sequence of embeddings and a $64{\times}64$ image diffusion model, followed by two super-resolution diffusion models for generating

---

[*]Equal contribution.

[†]Core contribution.

[‡]DF is also affiliated with the University of Toronto and the Vector Institute

36th Conference on Neural Information Processing Systems (NeurIPS 2022).

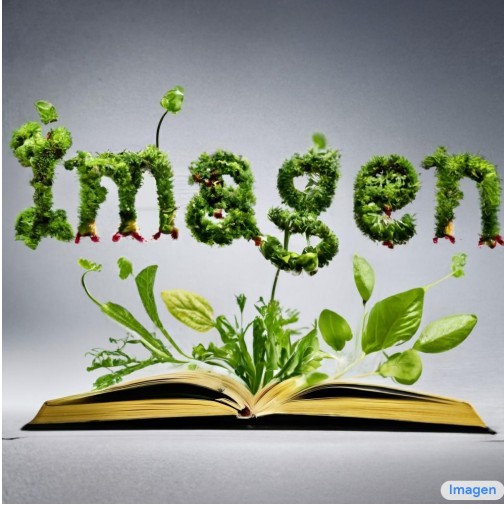

Sprouts in the shape of text 'Imagen' coming out of a fairytale book.

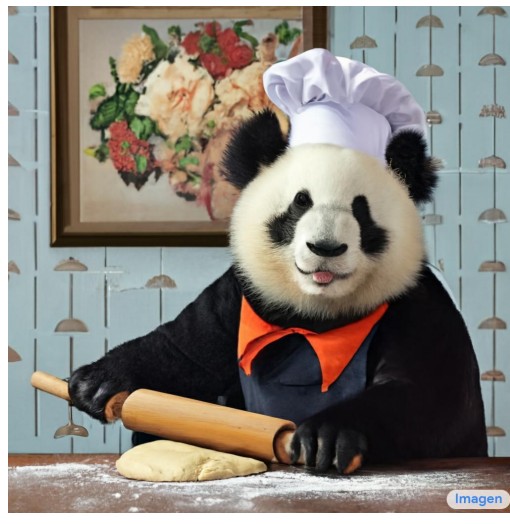

A photo of a Shiba Inu dog with a backpack riding a bike. It is wearing sunglasses and a beach hat.

A high contrast portrait of a very happy fuzzy panda dressed as a chef in a high end kitchen making dough. There is a painting of flowers on the wall behind him.

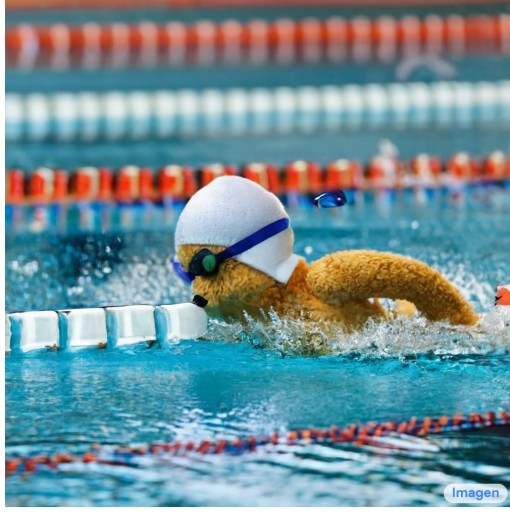

Teddy bears swimming at the Olympics 400m Butterfly event.

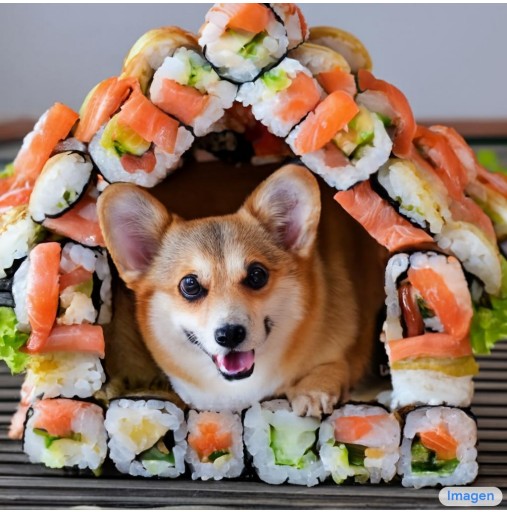

A cute corgi lives in a house made out of sushi.

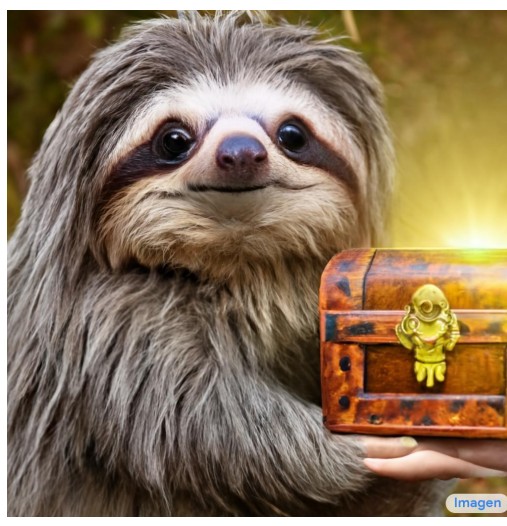

A cute sloth holding a small treasure chest. A bright golden glow is coming from the chest.

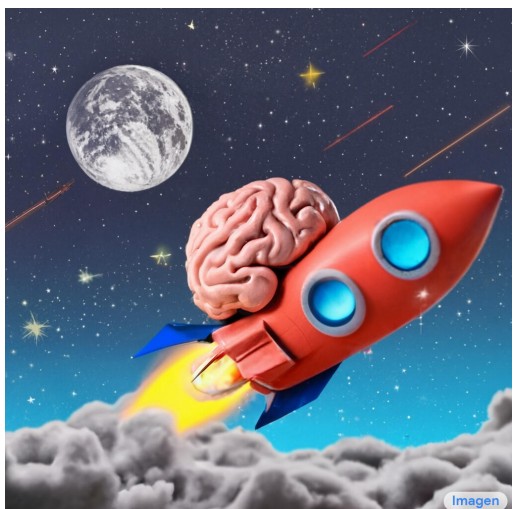

A brain riding a rocketship heading towards the moon.

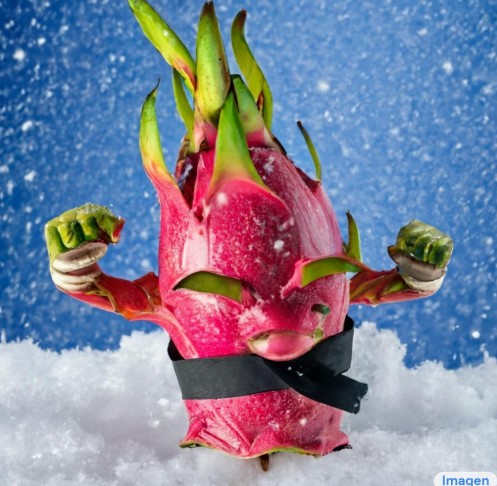

A dragon fruit wearing karate belt in the snow.

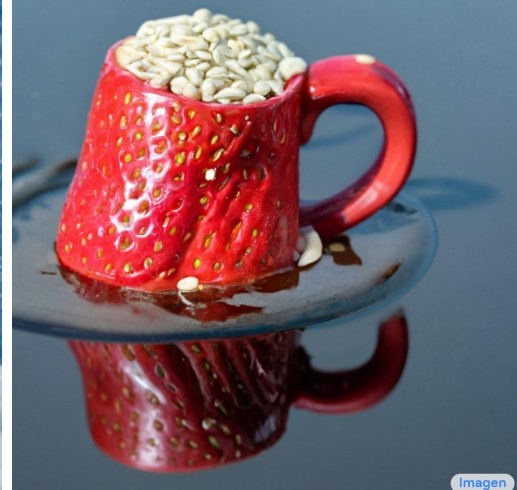

A strawberry mug filled with white sesame seeds. The mug is floating in a dark chocolate sea.

Figure 1: Select $1024 \times 1024$ Imagen samples for various text inputs. We only include photorealistic images in this figure and leave artistic content to the Appendix, since generating photorealistic images is more challenging from a technical point of view. Figs. A.1 to A.3 show more samples.

$256 \times 256$ and $1024 \times 1024$ images (see Fig. A.4). All diffusion models are conditioned on the text embedding sequence and use classifier-free guidance [27]. Imagen relies on new sampling techniques to allow usage of large guidance weights without sample quality degradation observed in prior work, resulting in images with higher fidelity and better image-text alignment than previously possible.

While conceptually simple and easy to train, Imagen yields surprisingly strong results. Imagen outperforms other methods on COCO [38] with zero-shot FID-30K of 7.27, significantly outperforming prior work such as GLIDE [43] (at 12.4) and the concurrent work of DALL-E 2 [56] (at 10.4). Our zero-shot FID score is also better than state-of-the-art models trained on COCO, e.g., Make-A-Scene [22] (at 7.6). Additionally, human raters indicate that generated samples from Imagen are on-par in image-text alignment to the reference images on COCO captions.

We introduce DrawBench, a new structured suite of text prompts for text-to-image evaluation. DrawBench enables deeper insights through a multi-dimensional evaluation of text-to-image models, with text prompts designed to probe different semantic properties of models. These include compositionality, cardinality, spatial relations, the ability to handle complex text prompts or prompts with rare words, and they include creative prompts that push the limits of models' ability to generate highly implausible scenes well beyond the scope of the training data. With DrawBench, extensive human evaluation shows that Imagen outperforms other recent methods [59, 12, 56] by a significant margin. We further demonstrate some of the clear advantages of the use of large pre-trained language models [54] over multi-modal embeddings such as CLIP [51] as a text encoder for Imagen.

Key contributions of the paper include:

1. We discover that large frozen language models trained only on text data are surprisingly very effective text encoders for text-to-image generation, and that scaling the size of frozen text encoder improves sample quality significantly more than scaling the size of image diffusion model.
2. We introduce *dynamic thresholding*, a new diffusion sampling technique to leverage high guidance weights and generating more photorealistic and detailed images than previously possible.
3. We highlight several important diffusion architecture design choices and propose *Efficient U-Net*, a new architecture variant which is simpler, converges faster and is more memory efficient.
4. We achieve a new state-of-the-art COCO FID of 7.27. Human raters find Imagen to be on-par with the reference images in terms of image-text alignment.
5. We introduce DrawBench, a new comprehensive and challenging evaluation benchmark for the text-to-image task. On DrawBench human evaluation, we find Imagen to outperform all other work, including the concurrent work of DALL-E 2 [56].

## 2   Imagen

Imagen consists of a text encoder that maps text to a sequence of embeddings and a cascade of conditional diffusion models that map these embeddings to images of increasing resolutions (see Fig. A.4). In the following subsections, we describe each of these components in detail.

### 2.1   Pretrained text encoders

Text-to-image models need powerful semantic text encoders to capture the complexity and compositionality of arbitrary natural language text inputs. Text encoders trained on paired image-text data are standard in current text-to-image models; they can be trained from scratch [43, 55] or pretrained on image-text data [56] (e.g., CLIP [51]). The image-text training objectives suggest that these text encoders may encode visually semantic and meaningful representations especially relevant for the text-to-image generation task. Large language models can be another models of choice to encode text for text-to-image generation. Recent progress in large language models (e.g., BERT [15], GPT [49, 50, 7], T5 [54]) have led to leaps in textual understanding and generative capabilities. Language models are trained on text only corpus significantly larger than paired image-text data, thus being exposed to a very rich and wide distribution of text. These models are also generally much larger than text encoders in current image-text models [51, 32, 83] (e.g. PaLM [11] has 540B parameters, while CoCa [83] has a $\approx$ 1B parameter text encoder).

It thus becomes natural to explore both families of text encoders for the text-to-image task. Imagen explores pretrained text encoders: BERT [15], T5 [53] and CLIP [48]. For simplicity, we freeze the weights of these text encoders. Freezing has several advantages such as offline computation of embeddings, resulting in negligible computation or memory footprint during training of the text-to-image model. In our work, we find that there is a clear conviction that scaling the text encoder size improves the quality of text-to-image generation. We also find that while T5-XXL and CLIP

text encoders perform similarly on simple benchmarks such as MS-COCO, human evaluators prefer T5-XXL encoders over CLIP text encoders in both image-text alignment and image fidelity on DrawBench, a set of challenging and compositional prompts. We refer the reader to Section 4.4 for summary of our findings, and Appendix D.1 for detailed ablations.

## 2.2 Diffusion models and classifier-free guidance

Here we give a brief introduction to diffusion models; a precise description is in Appendix A. Diffusion models [66, 28, 68] are a class of generative models that convert Gaussian noise into samples from a learned data distribution via an iterative denoising process. These models can be conditional, for example on class labels, text, or low-resolution images [e.g. 16, 29, 62, 61, 78, 43, 56]. A diffusion model $\hat{\mathbf{x}}_\theta$ is trained on a denoising objective of the form

$$\mathbb{E}_{\mathbf{x},\mathbf{c},\boldsymbol{\epsilon},t}\left[w_t\|\hat{\mathbf{x}}_\theta(\alpha_t\mathbf{x} + \sigma_t\boldsymbol{\epsilon}, \mathbf{c}) - \mathbf{x}\|_2^2\right] \tag{1}$$

where $(\mathbf{x},\mathbf{c})$ are data-conditioning pairs, $t \sim \mathcal{U}([0,1])$, $\boldsymbol{\epsilon} \sim \mathcal{N}(\mathbf{0},\mathbf{I})$, and $\alpha_t, \sigma_t, w_t$ are functions of $t$ that influence sample quality. Intuitively, $\hat{\mathbf{x}}_\theta$ is trained to denoise $\mathbf{z}_t := \alpha_t\mathbf{x} + \sigma_t\boldsymbol{\epsilon}$ into $\mathbf{x}$ using a squared error loss, weighted to emphasize certain values of $t$. Sampling such as the ancestral sampler [28] and DDIM [67] start from pure noise $\mathbf{z}_1 \sim \mathcal{N}(\mathbf{0},\mathbf{I})$ and iteratively generate points $\mathbf{z}_{t_1}, \ldots, \mathbf{z}_{t_T}$, where $1 = t_1 > \cdots > t_T = 0$, that gradually decrease in noise content. These points are functions of the $\mathbf{x}$-predictions $\hat{\mathbf{x}}_0^t := \hat{\mathbf{x}}_\theta(\mathbf{z}_t, \mathbf{c})$.

Classifier guidance [16] is a technique to improve sample quality while reducing diversity in conditional diffusion models using gradients from a pretrained model $p(\mathbf{c}|\mathbf{z}_t)$ during sampling. *Classifier-free guidance* [27] is an alternative technique that avoids this pretrained model by instead jointly training a single diffusion model on conditional and unconditional objectives via randomly dropping $\mathbf{c}$ during training (e.g. with 10% probability). Sampling is performed using the adjusted $\mathbf{x}$-prediction $(\mathbf{z}_t - \sigma\tilde{\boldsymbol{\epsilon}}_\theta)/\alpha_t$, where

$$\tilde{\boldsymbol{\epsilon}}_\theta(\mathbf{z}_t, \mathbf{c}) = w\boldsymbol{\epsilon}_\theta(\mathbf{z}_t, \mathbf{c}) + (1-w)\boldsymbol{\epsilon}_\theta(\mathbf{z}_t). \tag{2}$$

Here, $\boldsymbol{\epsilon}_\theta(\mathbf{z}_t, \mathbf{c})$ and $\boldsymbol{\epsilon}_\theta(\mathbf{z}_t)$ are conditional and unconditional $\boldsymbol{\epsilon}$-predictions, given by $\boldsymbol{\epsilon}_\theta := (\mathbf{z}_t - \alpha_t\hat{\mathbf{x}}_\theta)/\sigma_t$, and $w$ is the *guidance weight*. Setting $w = 1$ disables classifier-free guidance, while increasing $w > 1$ strengthens the effect of guidance. Imagen depends critically on classifier-free guidance for effective text conditioning.

## 2.3 Large guidance weight samplers

We corroborate the results of recent text-guided diffusion work [16, 43, 56] and find that increasing the classifier-free guidance weight improves image-text alignment, but damages image fidelity producing highly saturated and unnatural images [27]. We find that this is due to a train-test mismatch arising from high guidance weights. At each sampling step $t$, the $\mathbf{x}$-prediction $\hat{\mathbf{x}}_0^t$ must be within the same bounds as training data $\mathbf{x}$, i.e. within $[-1, 1]$, but we find empirically that high guidance weights cause $\mathbf{x}$-predictions to exceed these bounds. This is a train-test mismatch, and since the diffusion model is iteratively applied on its own output throughout sampling, the sampling process produces unnatural images and sometimes even diverges. To counter this problem, we investigate *static thresholding* and *dynamic thresholding*. See Appendix Fig. A.31 for reference implementation of the techniques and Appendix Fig. A.9 for visualizations of their effects.

**Static thresholding**: We refer to elementwise clipping the $\mathbf{x}$-prediction to $[-1, 1]$ as *static thresholding*. This method was in fact used but not emphasized in previous work [28], and to our knowledge its importance has not been investigated in the context of guided sampling. We discover that static thresholding is essential to sampling with large guidance weights and prevents generation of blank images. Nonetheless, static thresholding still results in over-saturated and less detailed images as the guidance weight further increases.

**Dynamic thresholding**: We introduce a new *dynamic thresholding* method: at each sampling step we set $s$ to a certain percentile absolute pixel value in $\hat{\mathbf{x}}_0^t$, and if $s > 1$, then we threshold $\hat{\mathbf{x}}_0^t$ to the range $[-s, s]$ and then divide by $s$. Dynamic thresholding pushes saturated pixels (those near -1 and 1) inwards, thereby actively preventing pixels from saturation at each step. We find that dynamic thresholding results in significantly better photorealism as well as better image-text alignment, especially when using very large guidance weights.

## 2.4 Robust cascaded diffusion models

Imagen utilizes a pipeline of a base $64 \times 64$ model, and two text-conditional super-resolution diffusion models to upsample a $64 \times 64$ generated image into a $256 \times 256$ image, and then to $1024 \times 1024$ image. Cascaded diffusion models with noise conditioning augmentation [29] have been extremely effective in progressively generating high-fidelity images. Furthermore, making the super-resolution models aware of the amount of noise added, via noise level conditioning, significantly improves the sample quality and helps improving the robustness of the super-resolution models to handle artifacts generated by lower resolution models [29]. Imagen uses noise conditioning augmentation for both the super-resolution models. We find this to be a critical for generating high fidelity images.

Given a conditioning low-resolution image and augmentation level (a.k.a aug_level) (e.g., strength of Gaussian noise or blur), we corrupt the low-resolution image with the augmentation (corresponding to aug_level), and condition the diffusion model on aug_level. During training, aug_level is chosen randomly, while during inference, we sweep over its different values to find the best sample quality. In our case, we use Gaussian noise as a form of augmentation, and apply variance preserving Gaussian noise augmentation resembling the forward process used in diffusion models (Appendix A). The augmentation level is specified using aug_level $\in [0, 1]$. See Fig. A.32 for reference pseudocode.

## 2.5 Neural network architecture

**Base model**: We adapt the U-Net [60] architecture from [42] for our base $64 \times 64$ text-to-image diffusion model. The network is conditioned on text embeddings via a pooled embedding vector, added to the diffusion timestep embedding similar to the class embedding conditioning method used in [16, 29]. We further condition on the entire sequence of text embeddings by adding cross attention [59] over the text embeddings at multiple resolutions. We study various methods of text conditioning in Appendix D.3.1. Furthermore, we found Layer Normalization [2] for text embeddings in the attention and pooling layers to help considerably improve performance.

**Super-resolution models**: For $64 \times 64 \rightarrow 256 \times 256$ super-resolution, we use the U-Net model adapted from [42, 61]. We make several modifications to this U-Net model for improving memory efficiency, inference time and convergence speed (our variant is 2-3x faster in steps/second over the U-Net used in [42, 61]). We call this variant *Efficient U-Net* (See Appendix B.1 for more details and comparisons). Our $256 \times 256 \rightarrow 1024 \times 1024$ super-resolution model trains on $64 \times 64 \rightarrow 256 \times 256$ crops of the $1024 \times 1024$ image. To facilitate this, we remove the self-attention layers, however we keep the text cross-attention layers which we found to be critical. During inference, the model receives the full $256 \times 256$ low-resolution images as inputs, and returns upsampled $1024 \times 1024$ images as outputs. Note that we use text cross attention for both our super-resolution models.

# 3 Evaluating Text-to-Image Models

The COCO [38] validation set is the standard benchmark for evaluating text-to-image models for both the supervised [85, 22] and the zero-shot setting [55, 43]. The key automated performance metrics used are FID [26] to measure image fidelity, and CLIP score [25, 51] to measure image-text alignment. Consistent with previous works, we report zero-shot FID-30K, for which 30K prompts are drawn randomly from the validation set, and the model samples generated on these prompts are compared with reference images from the full validation set. Since guidance weight is an important ingredient to control image quality and text alignment, we report most of our ablation results using trade-off (or *pareto*) curves between CLIP and FID scores across a range of guidance weights.

Both FID and CLIP scores have limitations, for example FID is not fully aligned with perceptual quality [44], and CLIP is ineffective at counting [51]. Due to these limitations, we use human evaluation to assess image quality and caption similarity, with ground truth reference caption-image pairs as a baseline. We use two experimental paradigms:

1. To probe image quality, the rater is asked to select between the model generation and reference image using the question: "Which image is more photorealistic (looks more real)?". We report the percentage of times raters choose model generations over reference images (the *preference rate*).
2. To probe alignment, human raters are shown an image and a prompt and asked "Does the caption accurately describe the above image?". They must respond with "yes", "somewhat", or "no". These responses are scored as 100, 50, and 0, respectively. These ratings are obtained independently for model samples and reference images, and both are reported.

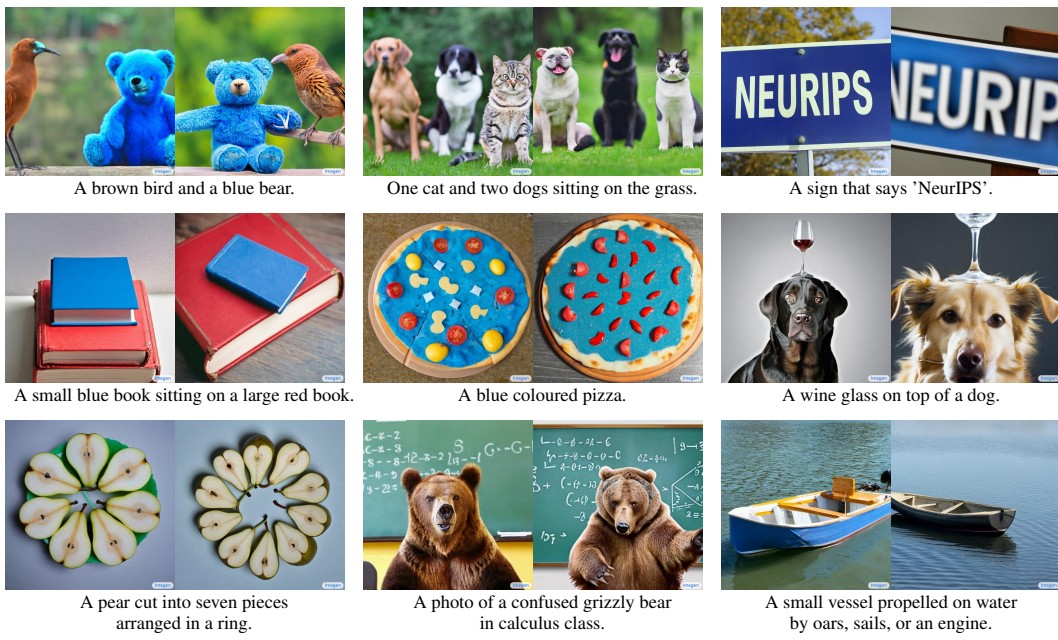

| | | |
|---|---|---|
| A brown bird and a blue bear. | One cat and two dogs sitting on the grass. | A sign that says 'NeurIPS'. |
| A small blue book sitting on a large red book. | A blue coloured pizza. | A wine glass on top of a dog. |
| A pear cut into seven pieces arranged in a ring. | A photo of a confused grizzly bear in calculus class. | A small vessel propelled on water by oars, sails, or an engine. |

Figure 2: Non-cherry picked Imagen samples for different categories of prompts from DrawBench.

For both cases we use 200 randomly chosen image-caption pairs from the COCO validation set. Subjects were shown batches of 50 images. We also used interleaved "control" trials, and only include rater data from those who correctly answered at least 80% of the control questions. This netted 73 and 51 ratings per image for image quality and image-text alignment evaluations, respectively.

**DrawBench**: While COCO is a valuable benchmark, it is increasingly clear that it has a limited spectrum of prompts that do not readily provide insight into differences between models (e.g., see Sec. 4.2). Recent work by [10] proposed a new evaluation set called PaintSkills to systematically evaluate visual reasoning skills and social biases beyond COCO. With similar motivation, we introduce *DrawBench*, a comprehensive and challenging set of prompts that support the evaluation and comparison of text-to-image models. DrawBench contains 11 categories of prompts, testing different capabilities of models such as the ability to faithfully render different colors, numbers of objects, spatial relations, text in the scene, and unusual interactions between objects. Categories also include complex prompts, including long, intricate textual descriptions, rare words, and also misspelled prompts. We also include sets of prompts collected from DALL-E [55], Gary Marcus et al. [40] and Reddit. Across these 11 categories, DrawBench comprises 200 prompts in total, striking a good balance between the desire for a large, comprehensive dataset, and small enough that human evaluation remains feasible. (Appendix C provides a more detailed description of DrawBench. Fig. 2 shows example prompts from DrawBench with Imagen samples.)

We use DrawBench to directly compare different models. To this end, human raters are presented with two sets of images, one from Model A and one from Model B, each of which has 8 samples. Human raters are asked to compare Model A and Model B on sample fidelity and image-text alignment. They respond with one of three choices: Prefer Model A; Indifferent; or Prefer Model B.

## 4 Experiments

Section 4.1 describes training details, Sections 4.2 and 4.3 analyze results on MS-COCO and DrawBench, and Section 4.4 summarizes our ablation studies and key findings. For all experiments below, the images are fair random samples from Imagen with no post-processing or re-ranking.

### 4.1 Training details

Unless specified, we train a 2B parameter model for the $64 \times 64$ text-to-image synthesis, and 600M and 400M parameter models for $64 \times 64 \rightarrow 256 \times 256$ and $256 \times 256 \rightarrow 1024 \times 1024$ for super-resolution respectively. We use a batch size of 2048 and 2.5M training steps for all models. We use 256 TPU-v4 chips for our base $64 \times 64$ model, and 128 TPU-v4 chips for both super-resolution

Table 1: MS-COCO $256 \times 256$ FID-30K. We use a guidance weight of 1.35 for our $64 \times 64$ model, and a guidance weight of 8.0 for our super-resolution model.

| Model | FID-30K | Zero-shot FID-30K |
|---|---|---|
| AttnGAN [79] | 35.49 | |
| DM-GAN [86] | 32.64 | |
| DF-GAN [72] | 21.42 | |
| DM-GAN + CL [81] | 20.79 | |
| XMC-GAN [84] | 9.33 | |
| LAFITE [85] | 8.12 | |
| Make-A-Scene [22] | 7.55 | |
| DALL-E [55] | | 17.89 |
| LAFITE [85] | | 26.94 |
| GLIDE [43] | | 12.24 |
| DALL-E 2 [56] | | 10.39 |
| **Imagen (Our Work)** | | **7.27** |

Table 2: COCO $256 \times 256$ human evaluation comparing model outputs and original images. For the bottom part (no people), we filter out prompts containing one of `man`, `men`, `woman`, `women`, `person`, `people`, `child`, `adult`, `adults`, `boy`, `boys`, `girl`, `girls`, `guy`, `lady`, `ladies`, `someone`, `toddler`, `(sport) player`, `workers`, `spectators`.

| Model | Photorealism ↑ | Alignment ↑ |
|---|---|---|
| *Original* | | |
| Original | 50.0% | $91.9 \pm 0.42$ |
| Imagen | $39.5 \pm 0.75\%$ | $91.4 \pm 0.44$ |
| *No people* | | |
| Original | 50.0% | $92.2 \pm 0.54$ |
| Imagen | $43.9 \pm 1.01\%$ | $92.1 \pm 0.55$ |

models. We do not find over-fitting to be an issue, and we believe further training might improve overall performance. We use Adafactor for our base $64 \times 64$ model, because initial comparisons with Adam suggested similar performance with much smaller memory footprint for Adafactor. For super-resolution models, we use Adam as we found Adafactor to hurt model quality in our initial ablations. For classifier-free guidance, we joint-train unconditionally via zeroing out the text embeddings with 10% probability for all three models. We train on a combination of internal datasets, with $\approx$ 460M image-text pairs, and the publicly available LAION-400M dataset [64], with $\approx$ 400M image-text pairs. There are limitations in our training data, and we refer the reader to Section 6 for details. See Appendix F for more implementation details.

## 4.2   Results on COCO

We evaluate Imagen on the COCO validation set using FID score, similar to [55, 43]. Table 1 displays the results. Imagen achieves state of the art *zero-shot* FID on COCO at 7.27, outperforming the concurrent work of DALL-E 2 [56] and even models trained on COCO. Table 2 reports the human evaluation to test image quality and alignment on the COCO validation set. We report results on the original COCO validation set, as well as a filtered version in which all reference data with people have been removed. For photorealism, Imagen achieves 39.2% preference rate indicating high image quality generation. On the set with no people, there is a boost in preference rate of Imagen to 43.6%, indicating Imagen's limited ability to generate photorealistic people. On caption similarity, Imagen's score is on-par with the original reference images, suggesting Imagen's ability to generate images that align well with COCO captions.

## 4.3   Results on DrawBench

Using DrawBench, we compare Imagen with DALL-E 2 (the public version) [56], GLIDE [43], Latent Diffusion [59], and CLIP-guided VQ-GAN [12]. Fig. 3 shows the human evaluation results for pairwise comparison of Imagen with each of the three models. We report the percentage of time raters prefer Model A, Model B, or are indifferent for both image fidelity and image-text alignment. We aggregate the scores across all the categories and raters. We find the human raters to exceedingly prefer Imagen over all others models in both image-text alignment and image fidelity. We refer the reader to Appendix E for a more detailed category wise comparison and qualitative comparison.

## 4.4   Analysis of Imagen

For a detailed analysis of Imagen see Appendix D. Key findings are discussed in Fig. 4 and below.

**Scaling text encoder size is extremely effective.** We observe that scaling the size of the text encoder leads to consistent improvement in both image-text alignment and image fidelity. Imagen trained with our largest text encoder, T5-XXL (4.6B parameters), yields the best results (Fig. 4a).

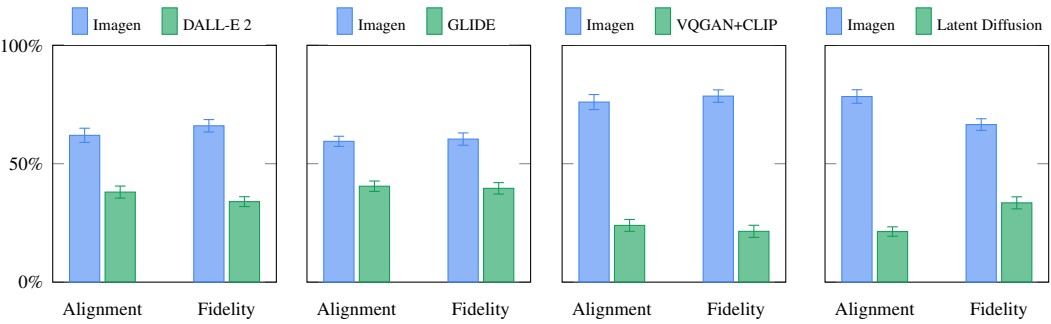

Figure 3: Comparison between Imagen and DALL-E 2 [56], GLIDE [43], VQ-GAN+CLIP [12] and Latent Diffusion [59] on DrawBench: User preference rates (with 95% confidence intervals) for image-text alignment and image fidelity.

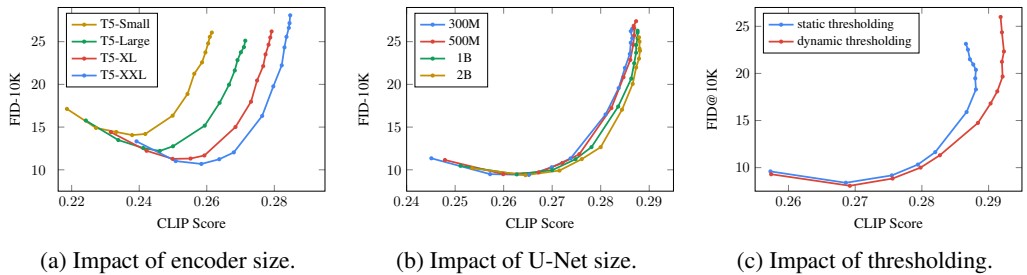

(a) Impact of encoder size.     (b) Impact of U-Net size.     (c) Impact of thresholding.

Figure 4: Summary of some of the critical findings of Imagen with pareto curves sweeping over different guidance values. See Appendix D for more details.

**Scaling text encoder size is more important than U-Net size.** While scaling the size of the diffusion model U-Net improves sample quality, we found scaling the text encoder size to be significantly more impactful than the U-Net size (Fig. 4b).

**Dynamic thresholding is critical.** We show that dynamic thresholding results in samples with significantly better photorealism and alignment with text, over static or no thresholding, especially under the presence of large classifier-free guidance weights (Fig. 4c).

**Human raters prefer T5-XXL over CLIP on DrawBench.** The models trained with T5-XXL and CLIP text encoders perform similarly on the COCO validation set in terms of CLIP and FID scores. However, we find that human raters prefer T5-XXL over CLIP on DrawBench across all 11 categories.

**Noise conditioning augmentation is critical.** We show that training the super-resolution models with noise conditioning augmentation leads to better CLIP and FID scores. We also show that noise conditioning augmentation enables stronger text conditioning for the super-resolution model, resulting in improved CLIP and FID scores at higher guidance weights. Adding noise to the low-res image during inference along with the use of large guidance weights allows the super-resolution models to generate diverse upsampled outputs while removing artifacts from the low-res image.

**Text conditioning method is critical.** We observe that conditioning over the sequence of text embeddings with cross attention significantly outperforms simple mean or attention based pooling in both sample fidelity as well as image-text alignment.

**Efficient U-Net is critical.** Our Efficient U-Net implementation uses less memory, converges faster, and has better sample quality with faster inference.

## 5   Related Work

Diffusion models have seen wide success in image generation [28, 42, 62, 16, 29, 61], outperforming GANs in fidelity and diversity, without training instability and mode collapse issues [6, 16, 29]. Autoregressive models [39], GANs [79, 84], VQ-VAE Transformer-based methods [55, 22], and diffusion models have seen remarkable progress in text-to-image [59, 43, 59], including the concurrent

DALL-E 2 [56], which uses a diffusion prior on CLIP text latents and cascaded diffusion models to generate high resolution $1024 \times 1024$ images; we believe Imagen is much simpler, as Imagen does not need to learn a latent prior, yet achieves better results in both MS-COCO FID and human evaluation on DrawBench. GLIDE [43] also uses cascaded diffusion models for text-to-image, but we use large pretrained frozen language models, which we found to be instrumental to both image fidelity and image-text alignment. XMC-GAN [84] also uses BERT as a text encoder, but we scale to much larger text encoders and demonstrate the effectiveness thereof. The use of cascaded models is also popular throughout the literature [14, 41] and has been used with success in diffusion models to generate high resolution images [16, 29].

# 6    Conclusions, Limitations and Societal Impact

Imagen showcases the effectiveness of frozen large pretrained language models as text encoders for the text-to-image generation using diffusion models. Our observation that scaling the size of these language models have significantly more impact than scaling the U-Net size on overall performance encourages future research directions on exploring even bigger language models as text encoders. Furthermore, through Imagen we re-emphasize the importance of classifier-free guidance, and we introduce dynamic thresholding, which allows usage of much higher guidance weights than seen in previous works. With these novel components, Imagen produces $1024 \times 1024$ samples with unprecedented photorealism and alignment with text.

Our primary aim with Imagen is to advance research on generative methods, using text-to-image synthesis as a test bed. While end-user applications of generative methods remain largely out of scope, we recognize the potential downstream applications of this research are varied and may impact society in complex ways. On the one hand, generative models have a great potential to complement, extend, and augment human creativity [30]. Text-to-image generation models, in particular, have the potential to extend image-editing capabilities and lead to the development of new tools for creative practitioners. On the other hand, generative methods can be leveraged for malicious purposes, including harassment and misinformation spread [20], and raise many concerns regarding social and cultural exclusion and bias [70, 65, 71]. These considerations inform our decision to not to release code or a public demo. In future work we will explore a framework for responsible externalization that balances the value of external auditing with the risks of unrestricted open-access.

Another ethical challenge relates to the large scale data requirements of text-to-image models, which have have led researchers to rely heavily on large, mostly uncurated, web-scraped datasets. While this approach has enabled rapid algorithmic advances in recent years, datasets of this nature have been critiqued and contested along various ethical dimensions. For example, public and academic discourse regarding appropriate use of public data has raised concerns regarding data subject awareness and consent [24, 18, 63, 45]. Dataset audits have revealed these datasets tend to reflect social stereotypes, oppressive viewpoints, and derogatory, or otherwise harmful, associations to marginalized identity groups [46, 4]. Training text-to-image models on this data risks reproducing these associations and causing significant representational harm that would disproportionately impact individuals and communities already experiencing marginalization, discrimination and exclusion within society. As such, there are a multitude of data challenges that must be addressed before text-to-image models like Imagen can be safely integrated into user-facing applications. While we do not directly address these challenges in this work, an awareness of the limitations of our training data guide our decision not to release Imagen for public use. We strongly caution against the use text-to-image generation methods for any user-facing tools without close care and attention to the contents of the training dataset.

Imagen's training data was drawn from several pre-existing datasets of image and English alt-text pairs. 400 million examples came from FIT400M, a cleaned version of the Alt-Text dataset [33, 31]. This data was filtered to removed noise and undesirable content, such as pornographic imagery and toxic language. However, a recent audit of another one of our data sources, LAION-400M [64], uncovered a wide range of inappropriate content including pornographic imagery, racist slurs, and harmful social stereotypes [4]. This finding informs our assessment that Imagen is not suitable for public use at this time and also demonstrates the value of rigorous dataset audits and comprehensive dataset documentation (e.g. [23, 47]) in informing consequent decisions about the model's appropriate and safe use. Imagen also relies on text encoders trained on uncurated web-scale data, and thus inherits the social biases and limitations of large language models [5, 3, 52].

While we leave an in-depth empirical analysis of social and cultural biases encoded by Imagen to future work, our small scale internal assessments reveal several limitations that guide our decision

not to release Imagen at this time. First, all generative models, including Imagen, Imagen, may run into danger of dropping modes of the data distribution, which may further compound the social consequence of dataset bias. Second, Imagen exhibits serious limitations when generating images depicting people. Our human evaluations found Imagen obtains significantly higher preference rates when evaluated on images that do not portray people, indicating a degradation in image fidelity. Finally, our preliminary assessment also suggests Imagen encodes several social biases and stereotypes, including an overall bias towards generating images of people with lighter skin tones and a tendency for images portraying different professions to align with Western gender stereotypes. Even when we focus generations away from people, our preliminary analysis indicates Imagen encodes a range of social and cultural biases when generating images of activities, events, and objects.

While there has been extensive work auditing image-to-text and image labeling models for forms of social bias (e.g. [8, 9, 71]), there has been comparatively less work on social bias evaluation methods for text-to-image models, with the recent exception of [10]. We believe this is a critical avenue for future research and we intend to explore benchmark evaluations for social and cultural bias in future work—for example, exploring whether it is possible to generalize the normalized pointwise mutual information metric [1] to the measurement of biases in image generation models. There is also a great need to develop a conceptual vocabulary around potential harms of text-to-image models that could guide the development of evaluation metrics and inform responsible model release. We aim to address these challenges in future work.

# 7    Acknowledgements

We give thanks to Ben Poole for reviewing our manuscript, early discussions, and providing many helpful comments and suggestions throughout the project. Special thanks to Kathy Meier-Hellstern, Austin Tarango, and Sarah Laszlo for helping us incorporate important responsible AI practices around this project. We appreciate valuable feedback and support from Elizabeth Adkison, Zoubin Ghahramani, Jeff Dean, Yonghui Wu, and Eli Collins. We are grateful to Tom Small for designing the Imagen watermark. We thank Jason Baldridge, Han Zhang, and Kevin Murphy for initial discussions and feedback. We acknowledge hard work and support from Fred Alcober, Hibaq Ali, Marian Croak, Aaron Donsbach, Tulsee Doshi, Toju Duke, Douglas Eck, Jason Freidenfelds, Brian Gabriel, Molly FitzMorris, David Ha, Philip Parham, Laura Pearce, Evan Rapoport, Lauren Skelly, Johnny Soraker, Negar Rostamzadeh, Vijay Vasudevan, Tris Warkentin, Jeremy Weinstein, and Hugh Williams for giving us advice along the project and assisting us with the publication process. We thank Victor Gomes and Erica Moreira for their consistent and critical help with TPU resource allocation. We also give thanks to Shekoofeh Azizi, Harris Chan, Chris A. Lee, and Nick Ma for volunteering a considerable amount of their time for testing out DrawBench. We thank Aditya Ramesh, Prafulla Dhariwal, and Alex Nichol for allowing us to use DALL-E 2 samples and providing us with GLIDE samples. We are thankful to Matthew Johnson and Roy Frostig for starting the JAX project and to the whole JAX team for building such a fantastic system for high-performance machine learning research. Special thanks to Durk Kingma, Jascha Sohl-Dickstein, Lucas Theis and the Toronto Brain team for helpful discussions and spending time Imagening!

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
