# Supplementary Material
## Photorealistic Text-to-Image Diffusion Models with Deep Language Understanding

Chitwan Saharia[1], William Chan[1], Saurabh Saxena[2], Lala Li[2], Jay Whang[2],
Emily Denton, Seyed Kamyar Seyed Ghasemipour, Burcu Karagol Ayan,
S. Sara Mahdavi, Raphael Gontijo-Lopes, Tim Salimans,
Jonathan Ho[2], David J Fleet[2], Mohammad Norouzi[1]

{sahariac,williamchan,mnorouzi}@google.com
{srbs,lala,jwhang,jonathanho,davidfleet}@google.com

Google Research, Brain Team
Toronto, Ontario, Canada

[1]Equal Contribution
[2]Core Contribution

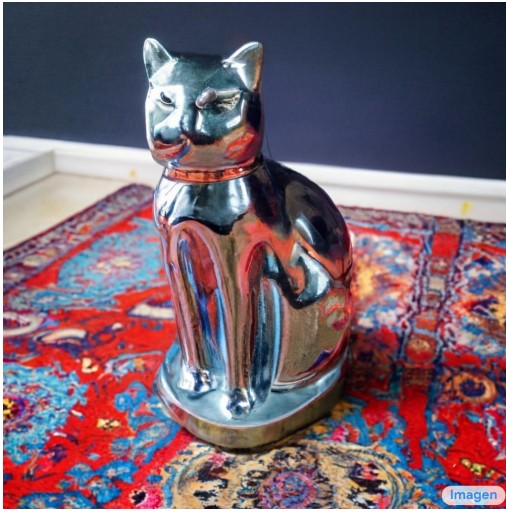
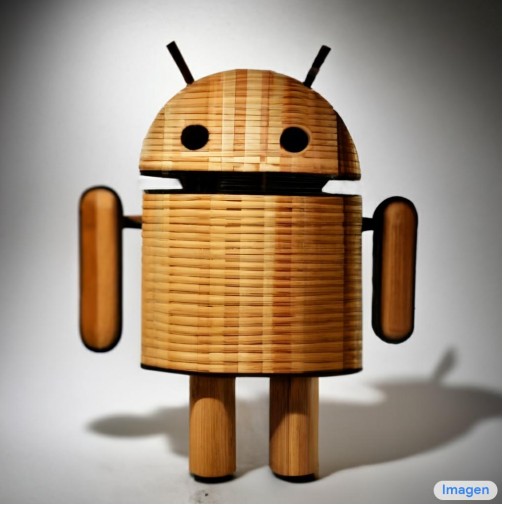
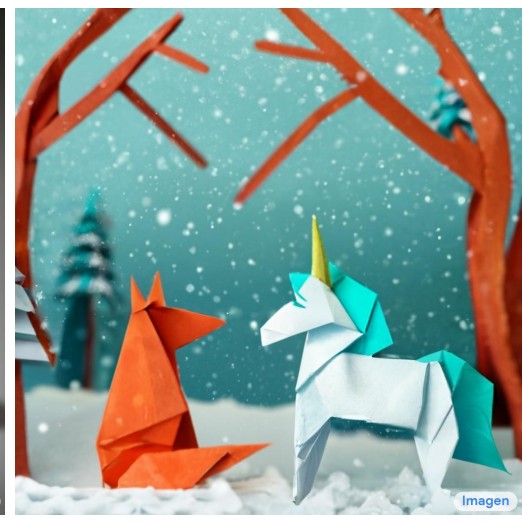

A chromeplated cat sculpture placed on a Persian rug.

Android Mascot made from bamboo.

Intricate origami of a fox and a unicorn in a snowy forest.

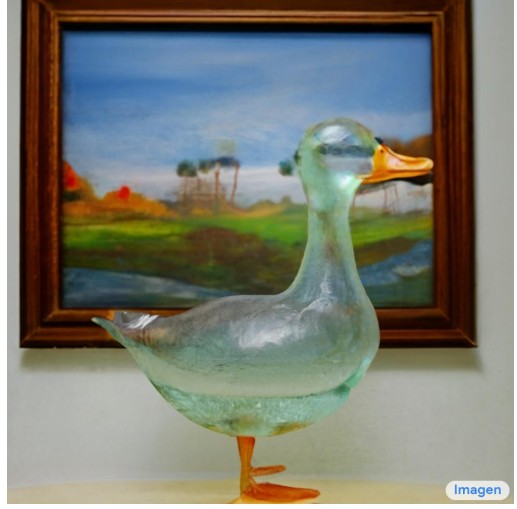
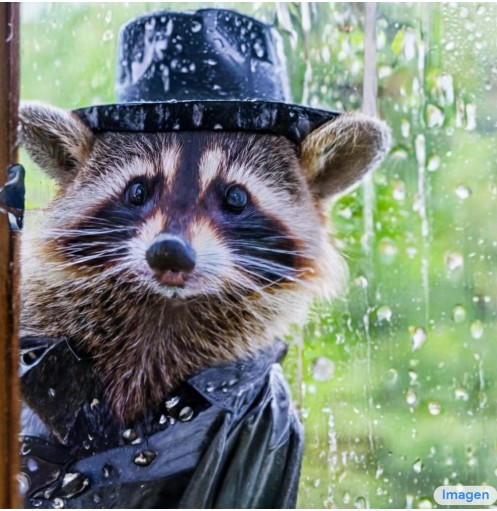
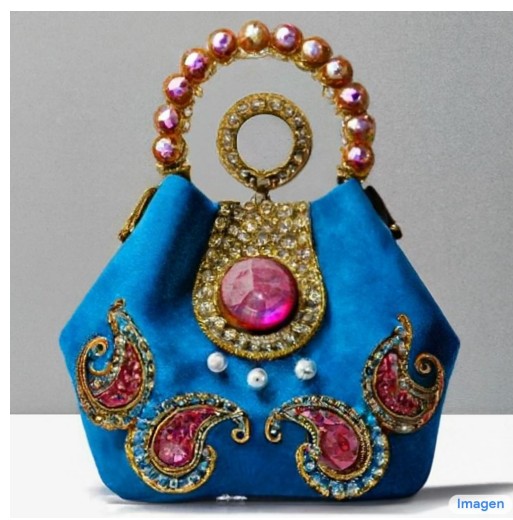

A transparent sculpture of a duck made out of glass.

A raccoon wearing cowboy hat and black leather jacket is behind the backyard window. Rain droplets on the window.

A bucket bag made of blue suede. The bag is decorated with intricate golden paisley patterns. The handle of the bag is made of rubies and pearls.

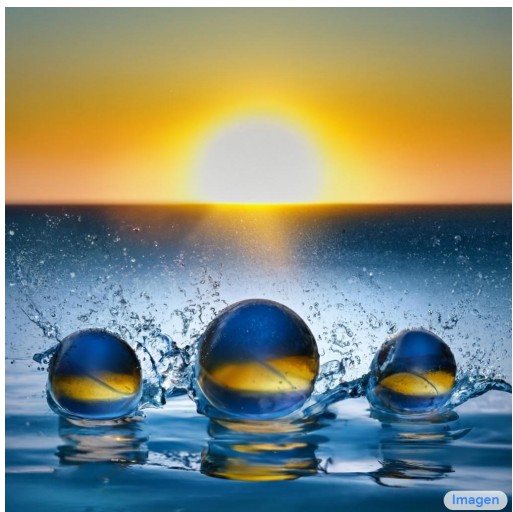
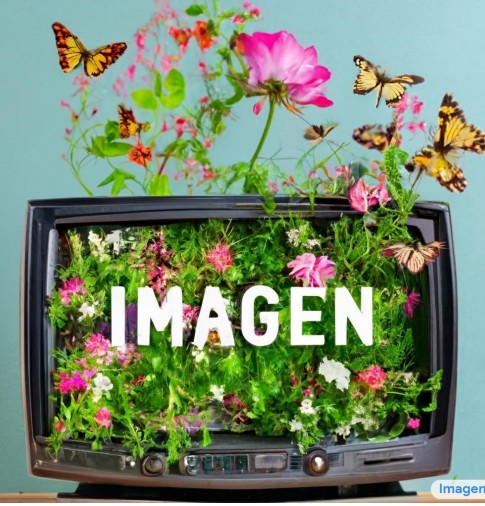
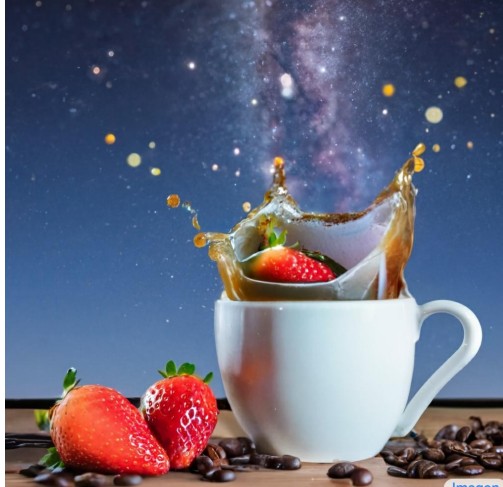

Three spheres made of glass falling into ocean. Water is splashing. Sun is setting.

Vines in the shape of text 'Imagen' with flowers and butterflies bursting out of an old TV.

A strawberry splashing in the coffee in a mug under the starry sky.

Figure A.1: Select $1024 \times 1024$ Imagen samples for various text inputs.

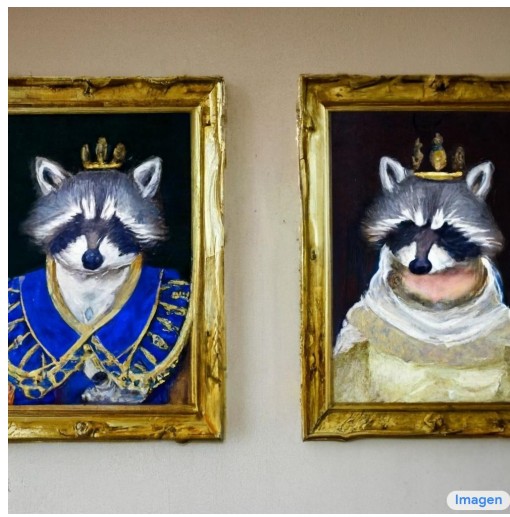 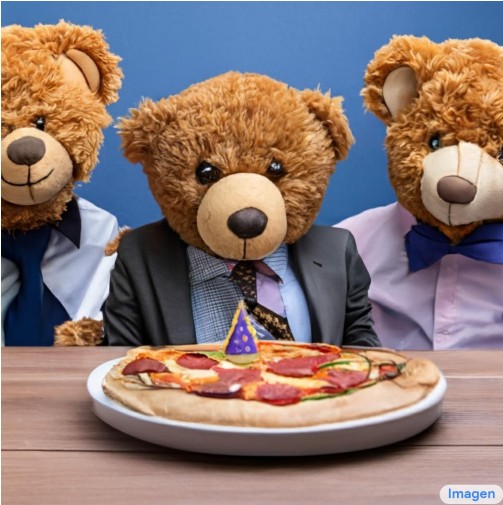 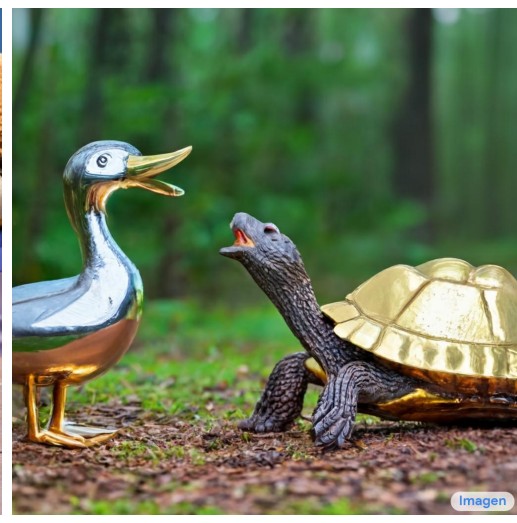

A wall in a royal castle. There are two paintings on the wall. The one on the left a detailed oil painting of the royal raccoon king. The one on the right a detailed oil painting of the royal raccoon queen.

A group of teddy bears in suit in a corporate office celebrating the birthday of their friend. There is a pizza cake on the desk.

A chrome-plated duck with a golden beak arguing with an angry turtle in a forest.

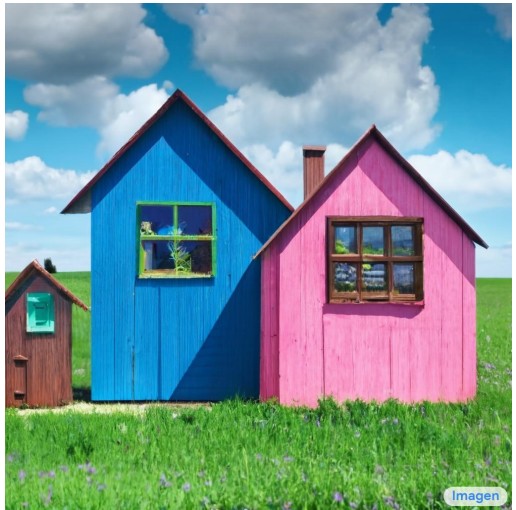 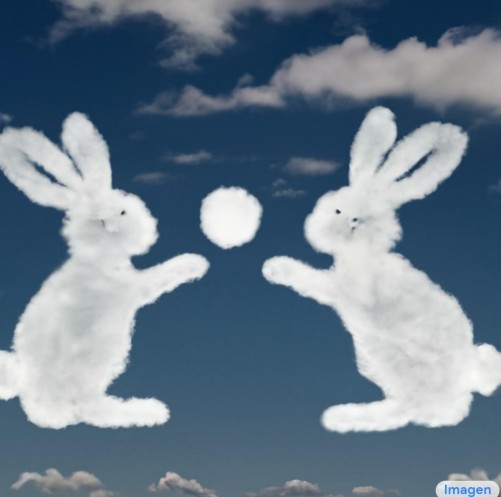 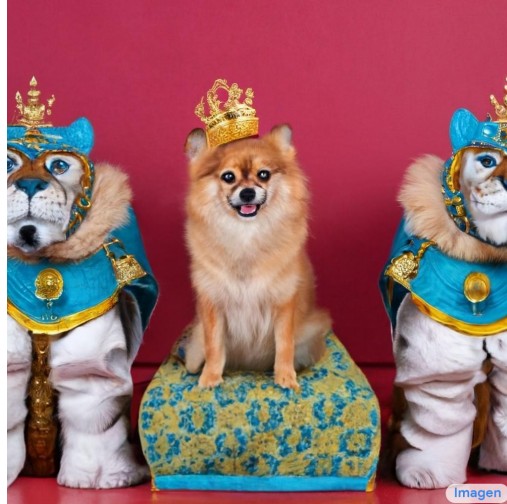

A family of three houses in a meadow. The Dad house is a large blue house. The Mom house is a large pink house. The Child house is a small wooden shed.

A cloud in the shape of two bunnies playing with a ball. The ball is made of clouds too.

A Pomeranian is sitting on the Kings throne wearing a crown. Two tiger soldiers are standing next to the throne.

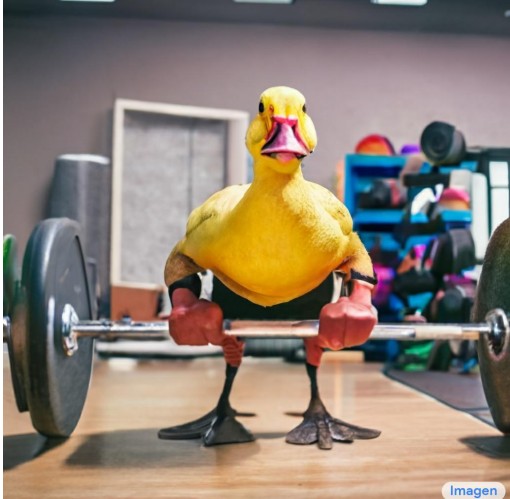 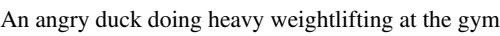 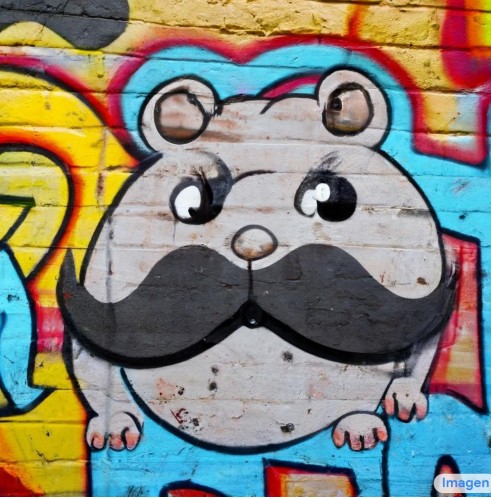 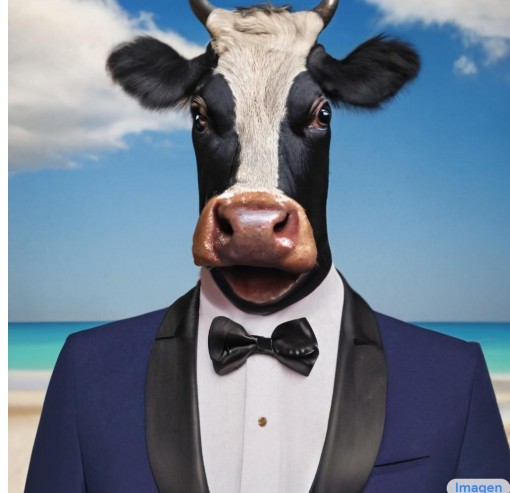

An angry duck doing heavy weightlifting at the gym.

A dslr picture of colorful graffiti showing a hamster with a moustache.

A photo of a person with the head of a cow, wearing a tuxedo and black bowtie. Beach wallpaper in the background.

Figure A.2: Select $1024 \times 1024$ Imagen samples for various text inputs.

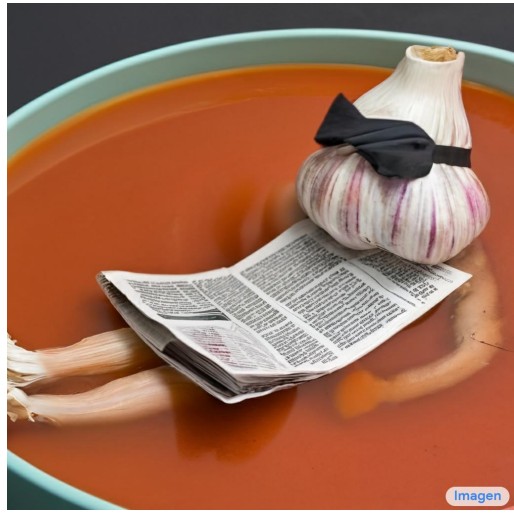
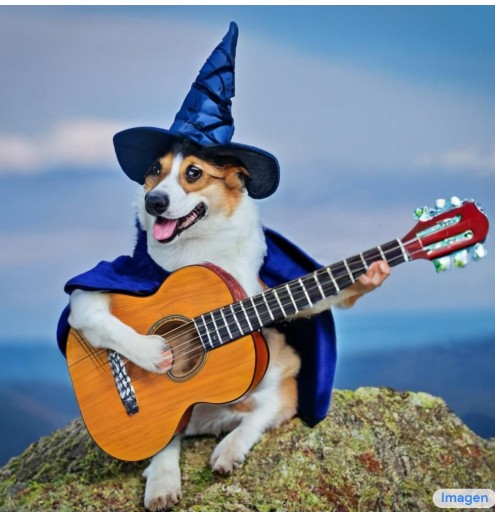
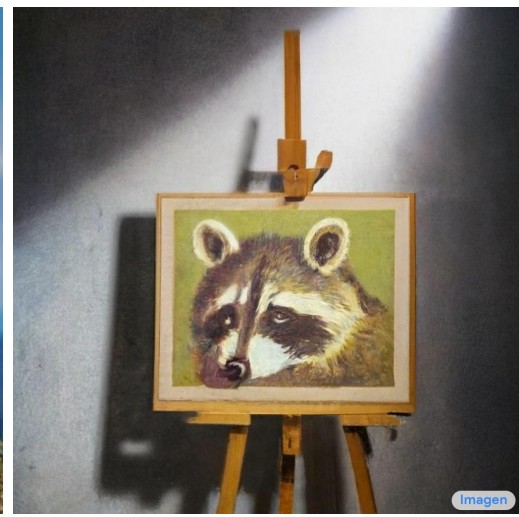

A relaxed garlic with a blindfold reading a newspaper while floating in a pool of tomato soup.

A photo of a corgi dog wearing a wizard hat playing guitar on the top of a mountain.

A single beam of light enter the room from the ceiling. The beam of light is illuminating an easel. On the easel there is a Rembrandt painting of a raccoon.

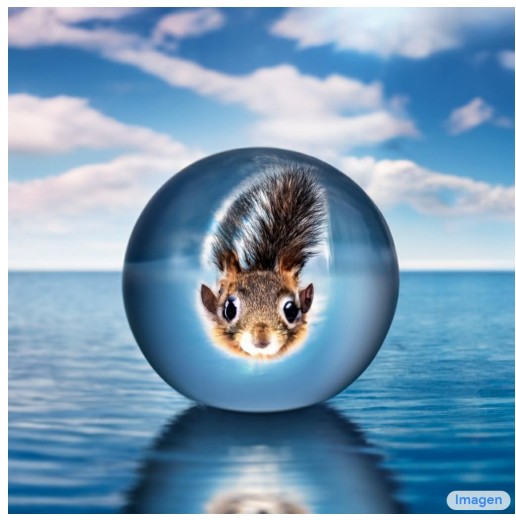
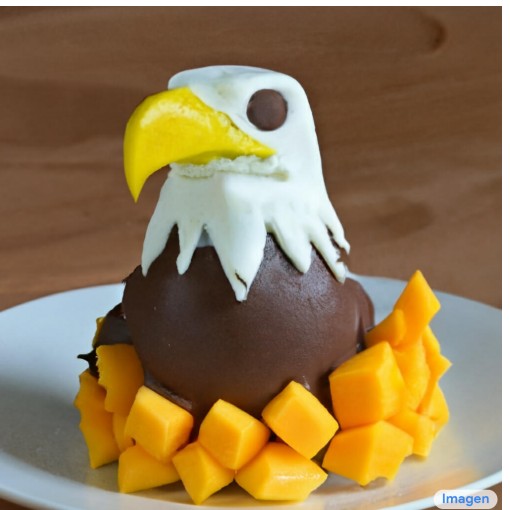
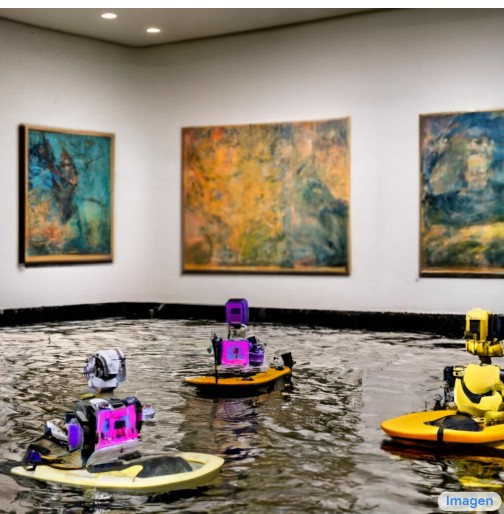

A squirrel is inside a giant bright shiny crystal ball on the surface of blue ocean. There are few clouds in the sky.

A bald eagle made of chocolate powder, mango, and whipped cream.

A marble statue of a Koala DJ in front of a marble statue of a turntable. The Koala has wearing large marble headphones.

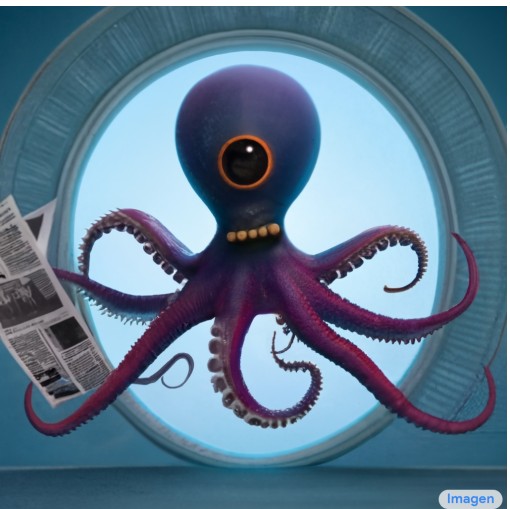
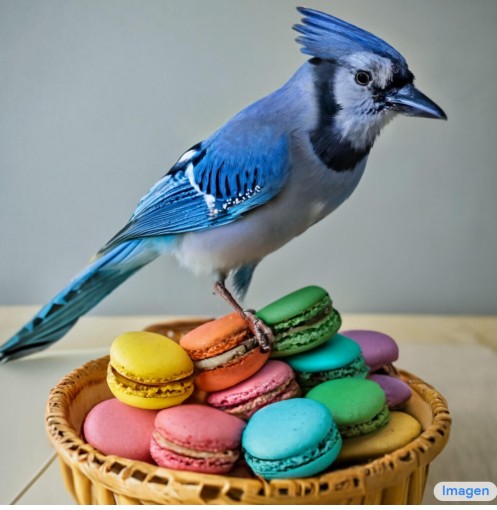

A photo of an alien octopus floats through a portal reading a newspaper.

A blue jay standing on a large basket of rainbow macarons.

An art gallery displaying Monet paintings. The art gallery is flooded. Robots are going around the art gallery using paddle boards.

Figure A.3: Select $1024 \times 1024$ Imagen samples for various text inputs.

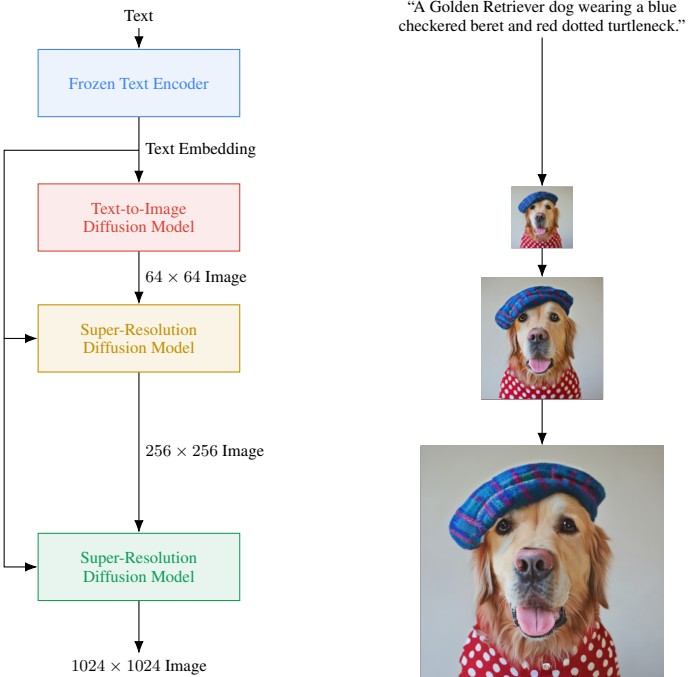

Figure A.4: Visualization of Imagen. Imagen uses a frozen text encoder to encode the input text into text embeddings. A conditional diffusion model maps the text embedding into a $64 \times 64$ image. Imagen further utilizes text-conditional super-resolution diffusion models to upsample the image, first $64 \times 64 \rightarrow 256 \times 256$, and then $256 \times 256 \rightarrow 1024 \times 1024$.

# A Background

Diffusion models are latent variable models with latents $\mathbf{z} = \{\mathbf{z}_t \,|\, t \in [0, 1]\}$ that obey a *forward process* $q(\mathbf{z}|\mathbf{x})$ starting at data $\mathbf{x} \sim p(\mathbf{x})$. This forward process is a Gaussian process that satisfies the Markovian structure:

$$q(\mathbf{z}_t|\mathbf{x}) = \mathcal{N}(\mathbf{z}_t; \alpha_t \mathbf{x}, \sigma_t^2 \mathbf{I}), \quad q(\mathbf{z}_t|\mathbf{z}_s) = \mathcal{N}(\mathbf{z}_t; (\alpha_t/\alpha_s)\mathbf{z}_s, \sigma_{t|s}^2 \mathbf{I}) \tag{3}$$

where $0 \leq s < t \leq 1$, $\sigma_{t|s}^2 = (1 - e^{\lambda_t - \lambda_s})\sigma_t^2$, and $\alpha_t, \sigma_t$ specify a differentiable *noise schedule* whose log signal-to-noise-ratio, i.e., $\lambda_t = \log[\alpha_t^2/\sigma_t^2]$, decreases with $t$ until $q(\mathbf{z}_1) \approx \mathcal{N}(\mathbf{0}, \mathbf{I})$. For generation, the diffusion model is learned to *reverse* this forward process.

Learning to reverse the forward process can be reduced to learning to denoise $\mathbf{z}_t \sim q(\mathbf{z}_t|\mathbf{x})$ into an estimate $\hat{\mathbf{x}}_\theta(\mathbf{z}_t, \lambda_t, \mathbf{c}) \approx \mathbf{x}$ for all $t$, where $\mathbf{c}$ is an optional conditioning signal (such as text embeddings or a low resolution image) drawn from the dataset jointly with $\mathbf{x}$. This is accomplished training $\hat{\mathbf{x}}_\theta$ using a weighted squared error loss

$$\mathbb{E}_{\boldsymbol{\epsilon}, t}\big[w(\lambda_t)\|\hat{\mathbf{x}}_\theta(\mathbf{z}_t, \lambda_t, \mathbf{c}) - \mathbf{x}\|_2^2\big] \tag{4}$$

where $t \sim \mathcal{U}([0, 1])$, $\boldsymbol{\epsilon} \sim \mathcal{N}(\mathbf{0}, \mathbf{I})$, and $\mathbf{z}_t = \alpha_t \mathbf{x} + \sigma_t \boldsymbol{\epsilon}$. This reduction of generation to denoising is justified as optimizing a weighted variational lower bound on the data log likelihood under the diffusion model, or as a form of denoising score matching [75, 68, 28, 37]. We use the $\boldsymbol{\epsilon}$-prediction parameterization, defined as $\hat{\mathbf{x}}_\theta(\mathbf{z}_t, \lambda_t, \mathbf{c}) = (\mathbf{z}_t - \sigma_t \boldsymbol{\epsilon}_\theta(\mathbf{z}_t, \lambda_t, \mathbf{c}))/\alpha_t$, and we impose a squared error loss on $\boldsymbol{\epsilon}_\theta$ in $\boldsymbol{\epsilon}$ space with $t$ sampled according to a cosine schedule [42]. This corresponds to a particular weighting $w(\lambda_t)$ and leads to a scaled score estimate $\boldsymbol{\epsilon}_\theta(\mathbf{z}_t, \lambda_t, \mathbf{c}) \approx -\sigma_t \nabla_{\mathbf{z}_t} \log p(\mathbf{z}_t|\mathbf{c})$, where $p(\mathbf{z}_t|\mathbf{c})$ is the true density of $\mathbf{z}_t$ given $\mathbf{c}$ under the forward process starting at $\mathbf{x} \sim p(\mathbf{x})$ [28, 37, 69]. Related model designs include the work of [73, 34, 35].

To sample from the diffusion model, we start at $\mathbf{z}_1 \sim \mathcal{N}(\mathbf{0}, \mathbf{I})$ and use the discrete time ancestral sampler [28] and DDIM [67] for certain models. DDIM follows the deterministic update rule

$$\mathbf{z}_s = \alpha_s \hat{\mathbf{x}}_\theta(\mathbf{z}_t, \lambda_t, \mathbf{c}) + \frac{\sigma_s}{\sigma_t}(\mathbf{z}_t - \alpha_t \hat{\mathbf{x}}_\theta(\mathbf{z}_t, \lambda_t, \mathbf{c})) \tag{5}$$

where $s < t$ follow a uniformly spaced sequence from 1 to 0. The ancestral sampler arises from a reversed description of the forward process; noting that $q(\mathbf{z}_s|\mathbf{z}_t, \mathbf{x}) = \mathcal{N}(\mathbf{z}_s; \tilde{\boldsymbol{\mu}}_{s|t}(\mathbf{z}_t, \mathbf{x}), \tilde{\sigma}_{s|t}^2 \mathbf{I})$, where $\tilde{\boldsymbol{\mu}}_{s|t}(\mathbf{z}_t, \mathbf{x}) = e^{\lambda_t - \lambda_s}(\alpha_s/\alpha_t)\mathbf{z}_t + (1 - e^{\lambda_t - \lambda_s})\alpha_s \mathbf{x}$ and $\tilde{\sigma}_{s|t}^2 = (1 - e^{\lambda_t - \lambda_s})\sigma_s^2$, it follows the stochastic update rule

$$\mathbf{z}_s = \tilde{\boldsymbol{\mu}}_{s|t}(\mathbf{z}_t, \hat{\mathbf{x}}_\theta(\mathbf{z}_t, \lambda_t, \mathbf{c})) + \sqrt{(\tilde{\sigma}_{s|t}^2)^{1-\gamma}(\sigma_{t|s}^2)^\gamma}\, \boldsymbol{\epsilon} \tag{6}$$

where $\boldsymbol{\epsilon} \sim \mathcal{N}(\mathbf{0}, \mathbf{I})$, and $\gamma$ controls the stochasticity of the sampler [42].

## B   Architecture Details

### B.1   Efficient U-Net

We introduce a new architectural variant, which we term Efficient U-Net, for our super-resolution models. We find our Efficient U-Net to be simpler, converges faster, and is more memory efficient compared to some prior implementations [42], especially for high resolutions. We make several key modifications to the U-Net architecture, such as shifting of model parameters from high resolution blocks to low resolution, scaling the skip connections by $1/\sqrt{2}$ similar to [69, 62] and reversing the order of downsampling/upsampling operations in order to improve the speed of the forward pass. Efficient U-Net makes several key modifications to the typical U-Net model used in [16, 61]:

- We shift the model parameters from the high resolution blocks to the low resolution blocks, via adding more residual blocks for the lower resolutions. Since lower resolution blocks typically have many more channels, this allows us to increase the model capacity through more model parameters, without egregious memory and computation costs.

- When using large number of residual blocks at lower-resolution (e.g. we use 8 residual blocks at lower-resolutions compared to typical 2-3 residual blocks used in standard U-Net architectures [16, 62]) we find that scaling the skip connections by $1/\sqrt{2}$ similar to [69, 62] significantly improves convergence speed.

- In a typical U-Net's downsampling block, the downsampling operation happens after the convolutions, and in an upsampling block, the upsampling operation happens prior the convolution. We reverse this order for both downsampling and upsampling blocks in order to significantly improve the speed of the forward pass of the U-Net, and find no performance degradation.

With these key simple modifications, Efficient U-Net is simpler, converges faster, and is more memory efficient compared to some prior U-Net implementations. Fig. A.30 shows the full architecture of Efficient U-Net, while Figures A.28 and A.29 show detailed description of the Downsampling and Upsampling blocks of Efficient U-Net respectively. See Appendix D.3.2 for results.

## C   DrawBench

In this section, we describe our new benchmark for fine-grained analysis of text-to-image models, namely, DrawBench. DrawBench consists of 11 categories with approximately 200 text prompts. This is large enough to test the model well, while small enough to easily perform trials with human raters. Table A.1 enumerates these categories along with description and few examples. We will release the full set of prompts in the camera ready version.

For evaluation on this benchmark, we conduct an independent human evaluation run for each category. For each prompt, the rater is shown two sets of images - one from Model A, and second from Model B. Each set contains 8 random (non-cherry

| Category | Description | Examples |
|---|---|---|
| Colors | Ability to generate objects with specified colors. | "A blue colored dog." 
 "A black apple and a green backpack." |
| Counting | Ability to generate specified number of objects. | "Three cats and one dog sitting on the grass." 
 "Five cars on the street." |
| Conflicting | Ability to generate conflicting interactions b/w objects. | "A horse riding an astronaut." 
 "A panda making latte art." |
| DALL-E [55] | Subset of challenging prompts from [55]. | "A triangular purple flower pot." 
 "A cross-section view of a brain." |
| Description | Ability to understand complex and long text prompts describing objects. | "A small vessel propelled on water by oars, sails, or an engine." 
 "A mechanical or electrical device for measuring time." |
| Marcus et al. [40] | Set of challenging prompts from [40]. | "A pear cut into seven pieces arranged in a ring." 
 "Paying for a quarter-sized pizza with a pizza-sized quarter." |
| Misspellings | Ability to understand misspelled prompts. | "Rbefraigerator." 
 "Tcennis rpacket." |
| Positional | Ability to generate objects with specified spatial positioning. | "A car on the left of a bus." 
 "A stop sign on the right of a refrigerator." |
| Rare Words | Ability to understand rare words[1]. | "Artophagous." 
 "Octothorpe." |
| Reddit | Set of challenging prompts from DALLE-2 Reddit[2]. | "A yellow and black bus cruising through the rainforest." 
 "A medieval painting of the wifi not working." |
| Text | Ability to generate quoted text. | "A storefront with 'Deep Learning' written on it." 
 "A sign that says 'Text to Image'." |

Table A.1: Description and examples of the 11 categories in DrawBench.

picked) generations from the corresponding model. The rater is asked two questions -

1. Which set of images is of higher quality?
2. Which set of images better represents the text caption : {Text Caption}?

where the questions are designed to measure: 1) image fidelity, and 2) image-text alignment. For each question, the rater is asked to select from three choices:

1. I prefer set A.
2. I am indifferent.
3. I prefer set B.

We aggregate scores from 25 raters for each category (totalling to $25 \times 11 = 275$ raters). We do not perform any post filtering of the data to identify unreliable raters, both for expedience and because the task was straightforward to explain and execute.

## D   Imagen Detailed Abalations and Analysis

In this section, we perform ablations and provide a detailed analysis of Imagen.

### D.1   Pre-trained Text Encoders

We explore several families of pre-trained text encoders: BERT [15], T5 [54], and CLIP [51]. There are several key differences between these encoders. BERT is trained on a smaller text-only corpus (approximately 20 GB, Wikipedia and BooksCorpus [87]) with a masking objective, and has relatively small model variants (upto 340M parameters). T5 is trained on a much larger C4 text-only corpus (approximately 800 GB) with a denoising objective, and has larger model variants

(up to 11B parameters). The CLIP model[3] is trained on an image-text corpus with an image-text contrastive objective. For T5 we use the encoder part for the contextual embeddings. For CLIP, we use the penultimate layer of the text encoder to get contextual embeddings. Note that we freeze the weights of these text encoders (i.e., we use off the shelf text encoders, without any fine-tuning on the text-to-image generation task). We explore a variety of model sizes for these text encoders.

We train a $64 \times 64$, 300M parameter diffusion model, conditioned on the text embeddings generated from BERT (base, and large), T5 (small, base, large, XL, and XXL), and CLIP (ViT-L/14). We observe that scaling the size of the language model text encoders generally results in better image-text alignment as captured by the CLIP score as a function of number of training steps (see Fig. A.6). One can see that the best CLIP scores are obtained with the T5-XXL text encoder.

Since guidance weights are used to control image quality and text alignment, we also report ablation results using curves that show the trade-off between CLIP and FID scores as a function of the guidance weights (see Fig. A.5a). We observe that larger variants of T5 encoder results in both better image-text alignment, and image fidelity. This emphasizes the effectiveness of large frozen text encoders for text-to-image models. Interestingly, we also observe that the T5-XXL encoder is on-par with the CLIP encoder when measured with CLIP and FID-10K on MS-COCO.

**T5-XXL vs CLIP on DrawBench**: We further compare T5-XXL and CLIP on DrawBench to perform a more comprehensive comparison of the abilities of these two text encoders. In our initial evaluations we observed that the 300M parameter models significantly underperformed on DrawBench. We believe this is primarily because DrawBench prompts are considerably more difficult than MS-COCO prompts.

In order to perform a meaningful comparison, we train $64 \times 64$ 1B parameter diffusion models with T5-XXL and CLIP text encoders for this evaluation. Fig. A.5b shows the results. We find that raters are considerably more likely to prefer the generations from the model trained with the T5-XXL encoder over the CLIP text encoder, especially for image-text alignment. This indicates that language models are better than text encoders trained on image-text contrastive objectives in encoding complex and compositional text prompts. Fig. A.7 shows the category specific comparison between the two models. We observe that human raters prefer T5-XXL samples over CLIP samples in all 11 categories for image-text alignment demonstrating the effectiveness of large language models as text encoders for text to image generation.

### D.2 Classifier-free Guidance and the Alignment-Fidelity Trade-off

We observe that classifier-free guidance [27] is a key contributor to generating samples with strong image-text alignment, this is also consistent with the observations of [55, 56]. There is typically a trade-off between image fidelity and image-text alignment, as we iterate over the guidance weight. While previous work

---

[3] https://github.com/openai/CLIP/blob/main/model-card.md

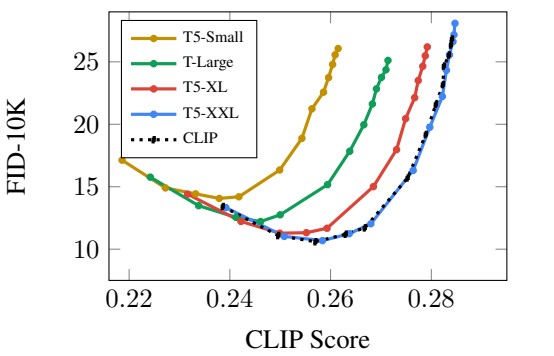

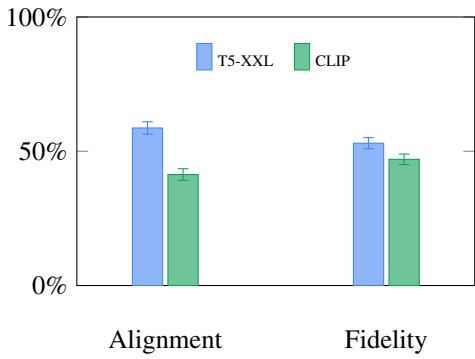

(a) Pareto curves comparing various text encoders.

(b) Comparing T5-XXL and CLIP on DrawBench.

Figure A.5: Comparison between text encoders for text-to-image generation. For Fig. A.5a, we sweep over guidance values of $[1, 1.25, 1.5, 1.75, 2, 3, 4, 5, 6, 7, 8, 9, 10]$

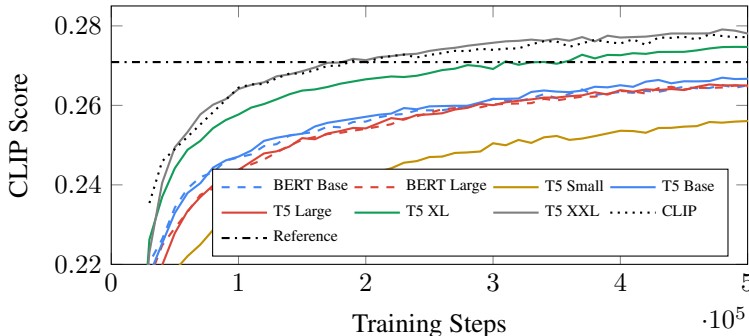

Figure A.6: Training convergence comparison between text encoders for text-to-image generation.

has typically used relatively small guidance weights, Imagen uses relatively large guidance weights for all three diffusion models. We found this to yield a good balance of sample quality and alignment. However, naive use of large guidance weights often produces relatively poor results. To enable the effective use of larger guidance we introduce several innovations, as described below.

**Thresholding Techniques**: First, we compare various thresholding methods used with classifier-free guidance. Fig. A.8 compares the CLIP vs. FID-10K score pareto frontiers for various thresholding methods of the base text-to-image $64 \times 64$ model. We observe that our dynamic thresholding technique results in significantly better CLIP scores, and comparable or better FID scores than the static thresholding technique for a wide range of guidance weights. Fig. A.9 shows qualitative samples for thresholding techniques.

**Guidance for Super-Resolution**: We further analyze the impact of classifier-free guidance for our $64 \times 64 \rightarrow 256 \times 256$ model. Fig. A.11a shows the pareto frontiers for CLIP vs. FID-10K score for the $64 \times 64 \rightarrow 256 \times 256$ super-resolution model. aug_level specifies the level of noise augmentation applied to the input low-resolution image during inference (aug_level $= 0$ means no noise). We observe that aug_level $= 0$ gives the best FID score for all values of guidance weight.

Furthermore, for all values of aug_level, we observe that FID improves considerably with increasing guidance weight upto around $7 - 10$. While generation

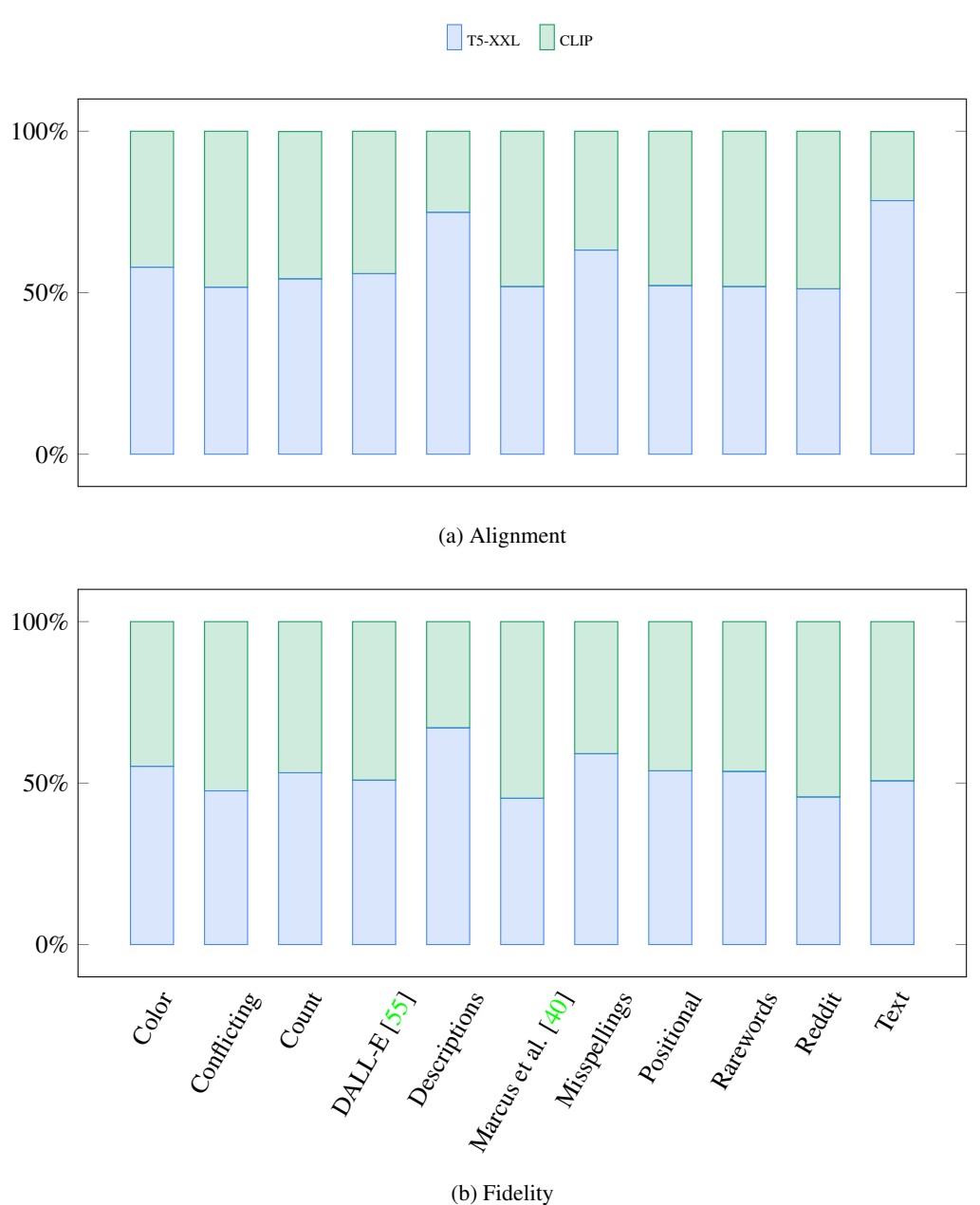

(a) Alignment

(b) Fidelity

Figure A.7: T5-XXL vs. CLIP text encoder on DrawBench a) image-text alignment, and b) image fidelity.

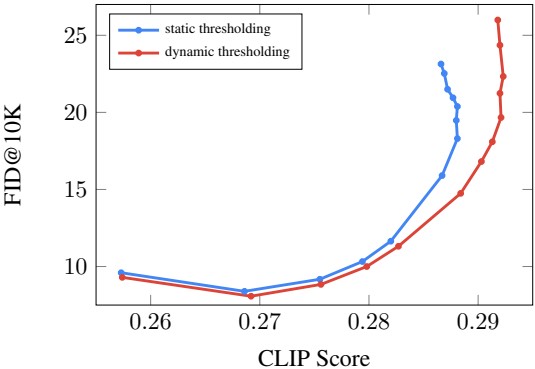

Figure A.8: CLIP Score vs FID trade-off across various $\hat{\mathbf{x}}_0$ thresholding methods for the $64{\times}64$ model. We sweep over guidance values of $[1, 1.25, 1.5, 1.75, 2, 3, 4, 5, 6, 7, 8, 9, 10]$.

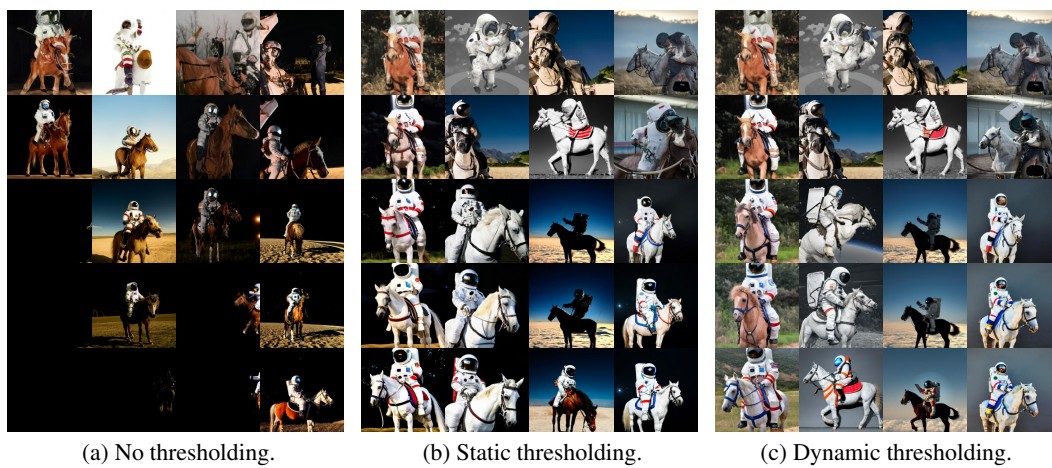

|                         |                          |                            |
| :---------------------: | :----------------------: | :------------------------: |
| (a) No thresholding.    | (b) Static thresholding. | (c) Dynamic thresholding.  |

Figure A.9: Thresholding techniques on $256 \times 256$ samples for "A photo of an astronaut riding a horse." Guidance weights increase from 1 to 5 as we go from top to bottom. No thresholding results in poor images with high guidance weights. Static thresholding is an improvement but still leads to oversaturated samples. Our dynamic thresholding leads to the highest quality images. See Fig. A.10 for more qualitative comparison.

using larger values of aug_level gives slightly worse FID, it allows more varied range of CLIP scores, suggesting more diverse generations by the super-resolution model. In practice, for our best samples, we generally use aug_level in $[0.1, 0.3]$. Using large values of aug_level and high guidance weights for the super-resolution models, Imagen can create different variations of a given $64 \times 64$ image by altering the prompts to the super-resolution models (See Fig. A.12 for examples).

**Impact of Conditioning Augmentation**: Fig. A.11b shows the impact of training super-resolution models with noise conditioning augmentation. Training with no noise augmentation generally results in worse CLIP and FID scores, suggesting noise conditioning augmentation is critical to attaining best sample quality similar to prior work [29]. Interestingly, the model trained without noise augmentation has much less variations in CLIP and FID scores across different guidance weights compared to the model trained with conditioning augmentation. We hypothesize that this is primarily because strong noise augmented training reduces the

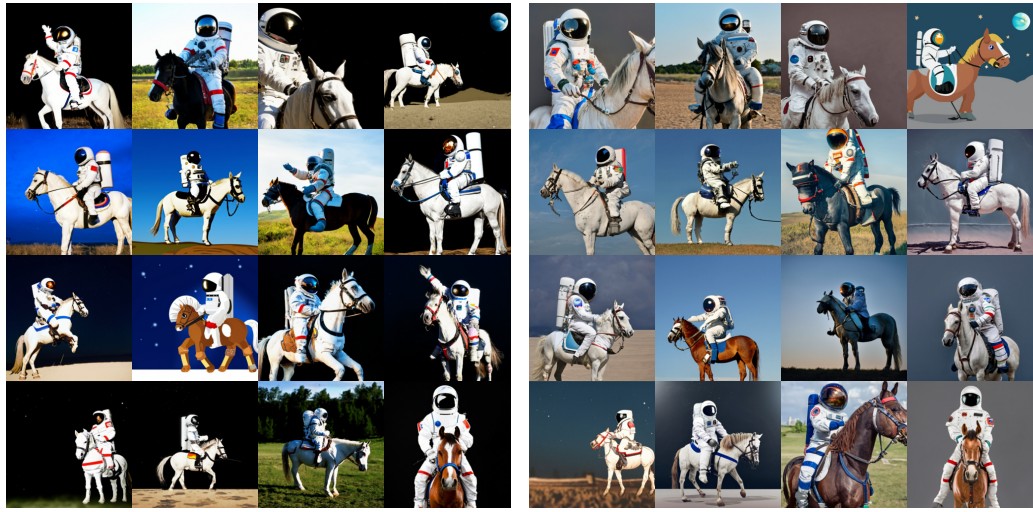

(a) Samples using static thresholding.  (b) Samples using dynamic thresholding ($p = 99.5$)

Figure A.10: Static vs. dynamic thresholding on non-cherry picked $256 \times 256$ samples using a guidance weight of 5 for both the base model and the super-resolution model, using the same random seed. The text prompt used for these samples is "A photo of an astronaut riding a horse." When using high guidance weights, static thresholding often leads to oversaturated samples, while our dynamic thresholding yields more natural looking images.

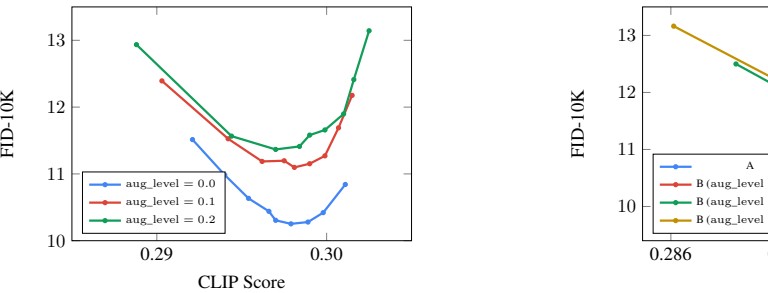

(a) Comparison between different values of aug_level. (b) Comparison between training with no noise augmentation "A" vs noise augmentation "B"

Figure A.11: CLIP vs FID-10K pareto curves showing the impact of noise augmentation on our $64 \times 64 \rightarrow 256 \times 256$ model. For each study, we sweep over guidance values of $[1, 3, 5, 7, 8, 10, 12, 15, 18]$

low-resolution image conditioning signal considerably, encouraging higher degree of dependence on conditioned text for the model.

### D.3 Impact of Model Size

Fig. A.13b plots the CLIP-FID score trade-off curves for various model sizes of the $64 \times 64$ text-to-image U-Net model. We train each of the models with a batch size of 2048, and 400K training steps. As we scale from 300M parameters to 2B parameters for the U-Net model, we obtain better trade-off curves with increasing model capacity. Interestingly, scaling the frozen text encoder model size yields more improvement in model quality over scaling the U-Net model size. Scaling with a frozen text encoder is also easier since the text embeddings can be computed and stored offline during training.

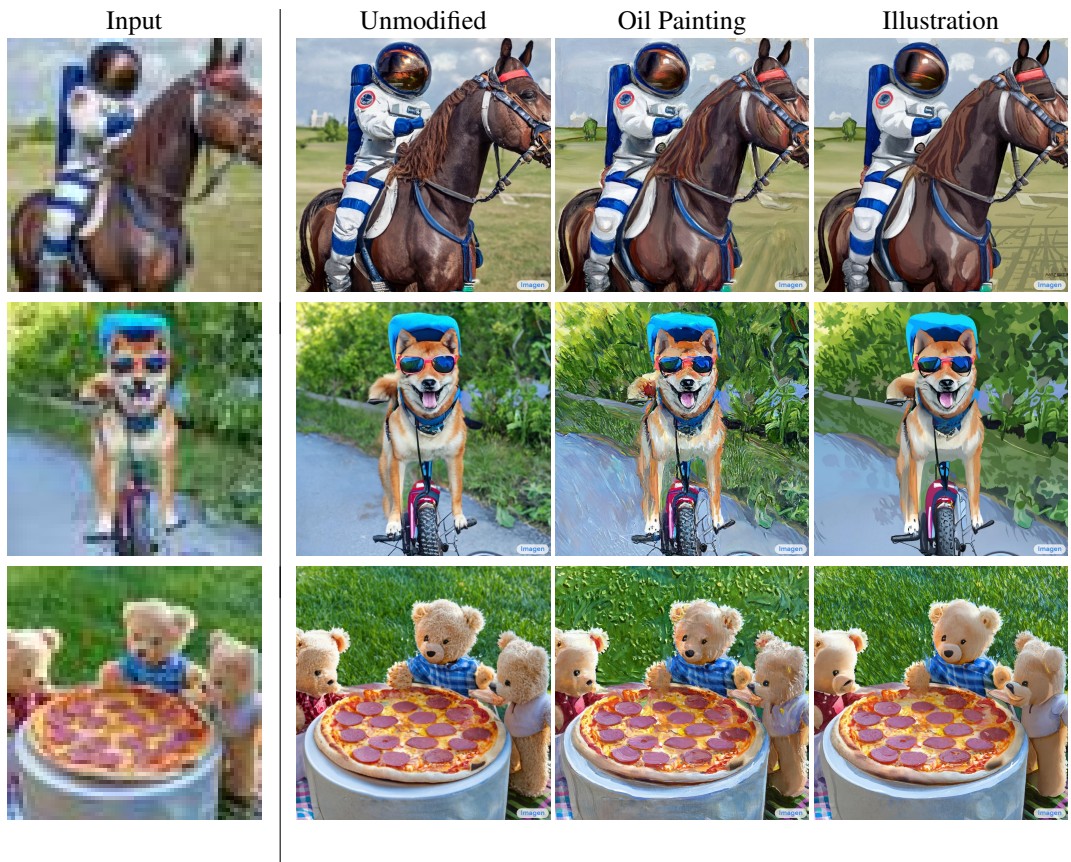

| Input | Unmodified | Oil Painting | Illustration |

Figure A.12: Super-resolution variations for some $64 \times 64$ generated images. We first generate the $64 \times 64$ image using "A photo of ... .". Given generated $64 \times 64$ images, we condition both the super-resolution models on different prompts in order to generate different upsampled variations. e.g. for oil painting we condition the super-resolution models on the prompt "An oil painting of ... .". Through a combination of large guidance weights and aug_level $= 0.3$ for both super-res models we can generate different styles based on the style query through text.

### D.3.1 Impact of Text Conditioning Schemas

We ablate various schemas for conditioning the frozen text embeddings in the base $64 \times 64$ text-to-image diffusion model. Fig. A.13a compares the CLIP-FID pareto curves for mean pooling, attention pooling, and cross attention. We find using any pooled embedding configuration (mean or attention pooling) performs noticeably worse compared to attending over the sequence of contextual embeddings in the attention layers. We implement the cross attention by concatenating the text embedding sequence to the key-value pairs of each self-attention layer in the base $64 \times 64$ and $64 \times 64 \rightarrow 256 \times 256$ models. For our $256 \times 256 \rightarrow 1024 \times 1024$ model, since we have no self-attention layers, we simply added explicit cross-attention layers to attend over the text embeddings. We found this to improve both fidelity and image-text alignment with minimal computational costs.

### D.3.2 Comparison of U-Net vs Efficient U-Net

We compare the performance of U-Net with our new Efficient U-Net on the task of $64 \times 64 \rightarrow 256 \times 256$ super-resolution task. Fig. A.14 compares the training

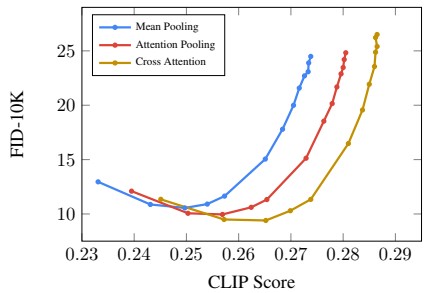
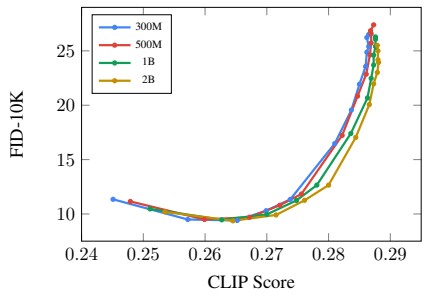

(a) Comparison between different text encoders.  (b) Comparison between different model sizes.

Figure A.13: CLIP vs FID-10K pareto curves for different ablation studies for the base $64 \times 64$ model. For each study, we sweep over guidance values of $[1, 1.25, 1.5, 1.75, 2, 3, 4, 5, 6, 7, 8, 9, 10]$

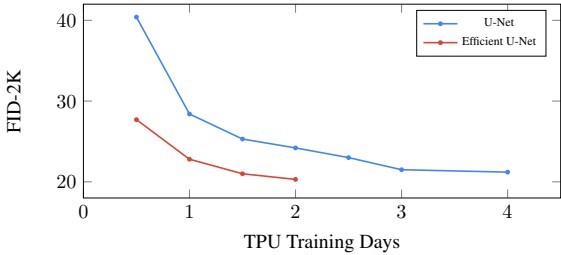

Figure A.14: Comparison of convergence speed of U-Net vs Efficient U-Net on the $64 \times 64 \rightarrow 256 \times 256$ super-resolution task.

convergence of the two architectures. We observe that Efficient U-Net converges significantly faster than U-Net, and obtains better performance overall. Our Efficient U-Net is also $\times 2 - 3$ faster at sampling.

# E   Comparison to GLIDE and DALL-E 2

Fig. A.15 shows category wise comparison between Imagen and DALL-E 2 [56] on DrawBench. We observe that human raters clearly prefer Imagen over DALL-E 2 in 7 out of 11 categories for text alignment. For sample fidelity, they prefer Imagen over DALL-E 2 in all 11 categories. Figures A.17 to A.21 show few qualitative comparisons between Imagen and DALL-E 2 samples used for this human evaluation study. Some of the categories where Imagen has a considerably larger preference over DALL-E 2 include Colors, Positional, Text, DALL-E and Descriptions. The authors in [56] identify some of these limitations of DALL-E 2, specifically they observe that DALLE-E 2 is worse than GLIDE [43] in binding attributes to objects such as colors, and producing coherent text from the input prompt (cf. the discussion of limitations in [56]). To this end, we also perform quantitative and qualitative comparison with GLIDE [43] on DrawBench. See Fig. A.16 for category wise human evaluation comparison between Imagen and GLIDE. See Figures A.22 to A.26 for qualitative comparisons. Imagen outperforms GLIDE on 8 out of 11 categories on image-text alignment, and 10 out of 11 categories on image fidelity. We observe that GLIDE is considerably better than DALL-E 2 in binding attributes to objects corroborating the observation by [56].

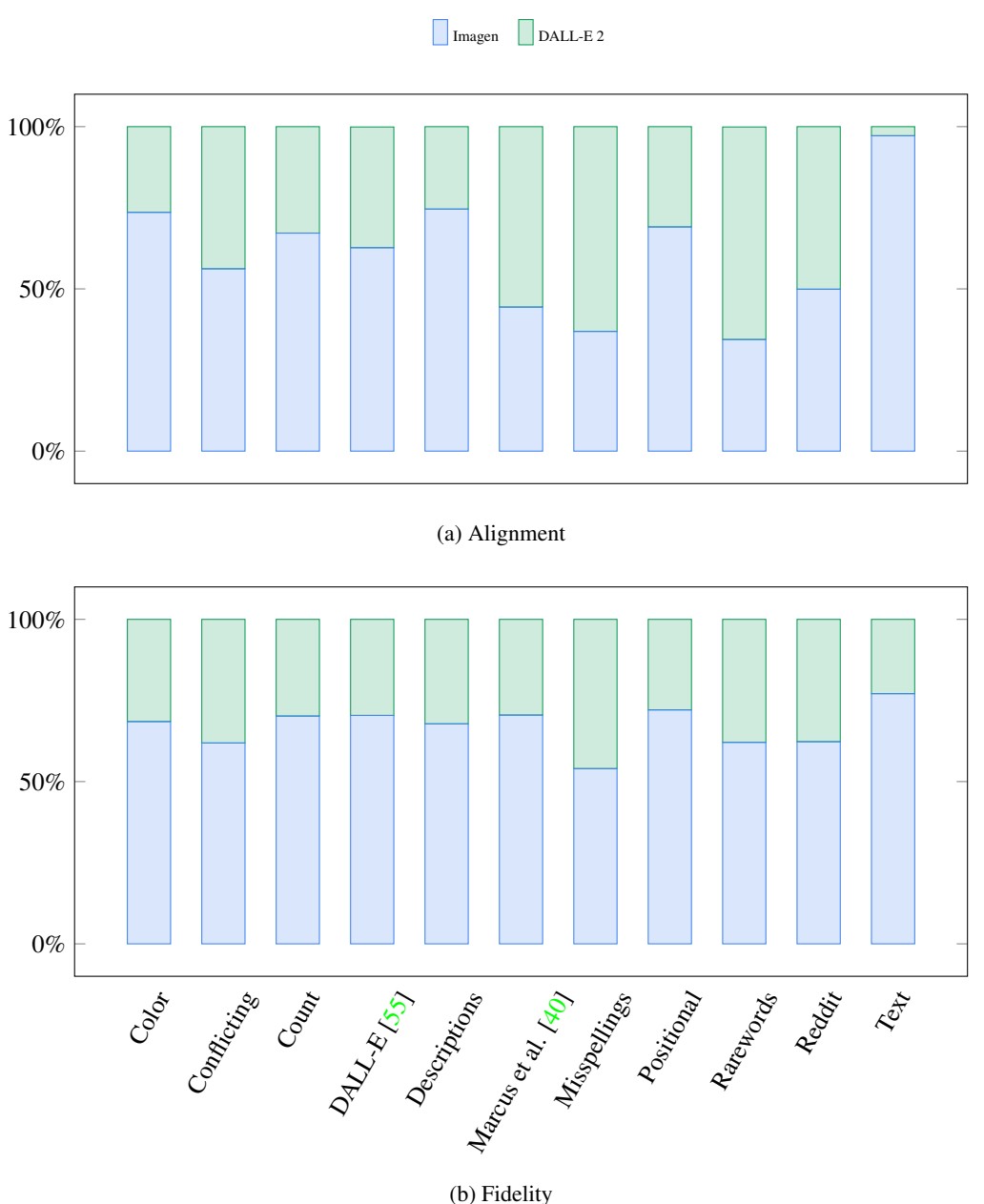

(a) Alignment

(b) Fidelity

Figure A.15: Imagen vs DALL-E 2 on DrawBench a) image-text alignment, and b) image fidelity.

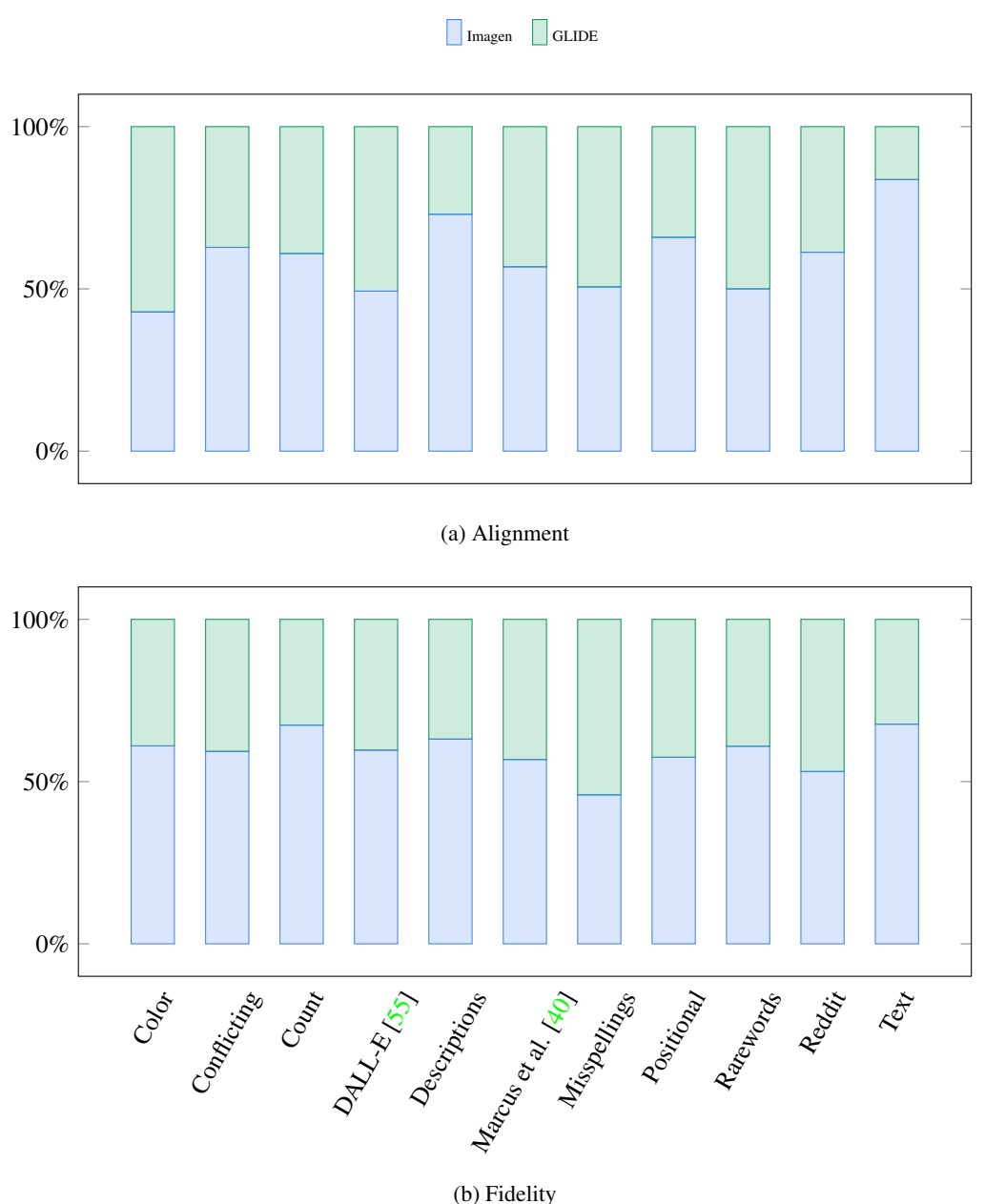

(a) Alignment

(b) Fidelity

Figure A.16: Imagen vs GLIDE on DrawBench a) image-text alignment, and b) image fidelity.

Imagen (Ours)                                    DALL-E 2 [56]

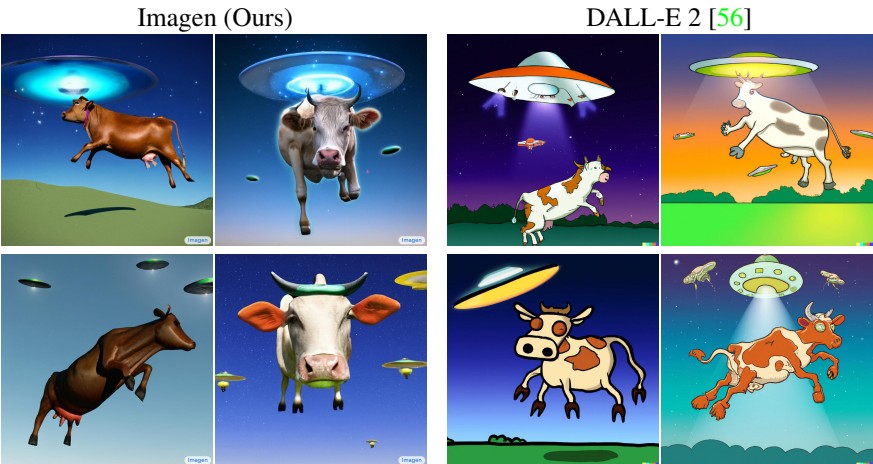

Hovering cow abducting aliens.

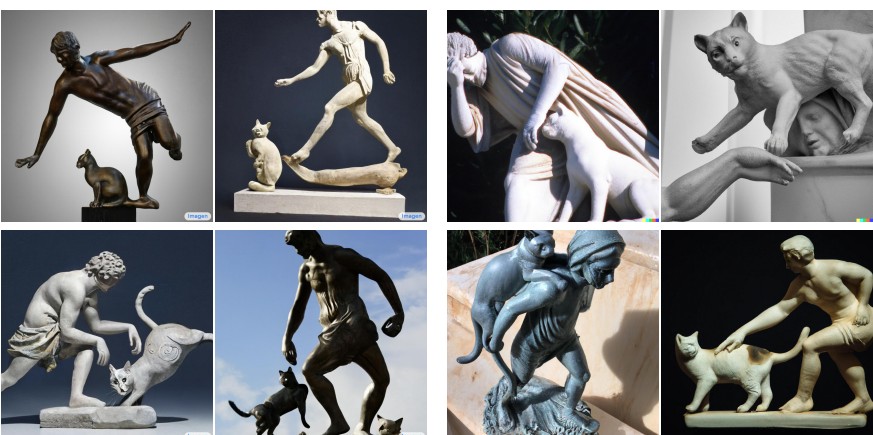

Greek statue of a man tripping over a cat.

Figure A.17: Example qualitative comparisons between Imagen and DALL-E 2 [56] on DrawBench prompts from Reddit category.

Imagen (Ours)  DALL-E 2 [56]

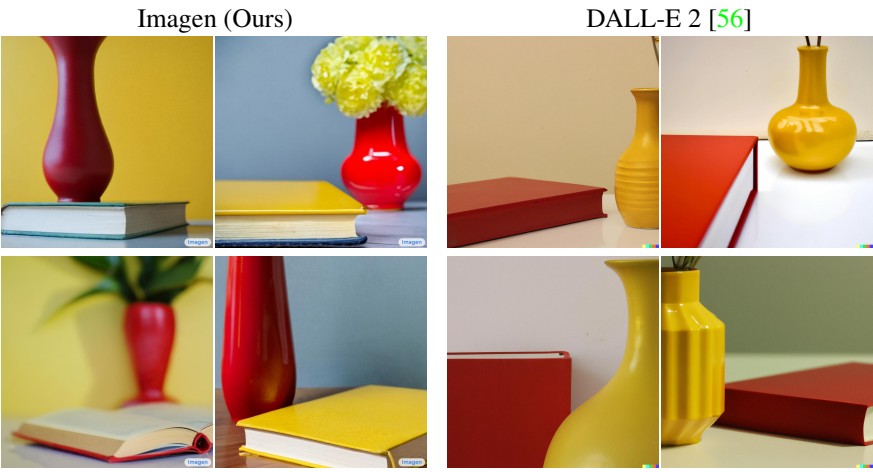

A yellow book and a red vase.

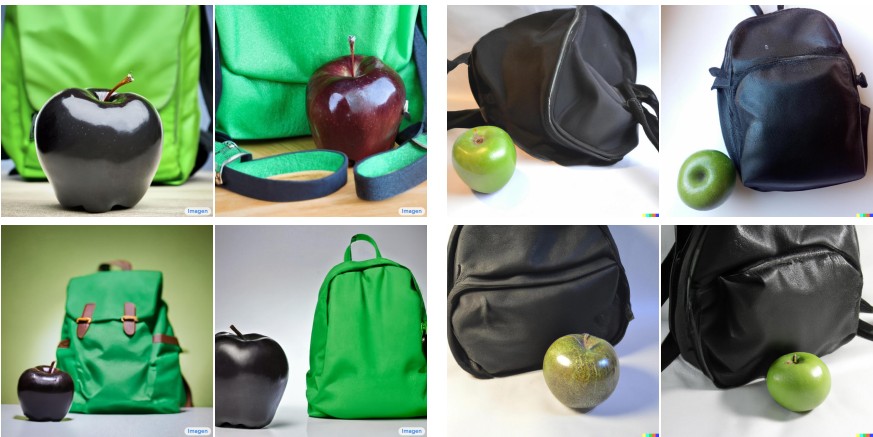

A black apple and a green backpack.

Figure A.18: Example qualitative comparisons between Imagen and DALL-E 2 [56] on DrawBench prompts from Colors category. We observe that DALL-E 2 generally struggles with correctly assigning the colors to the objects especially for prompts with more than one object.

Imagen (Ours) | DALL-E 2 [56]

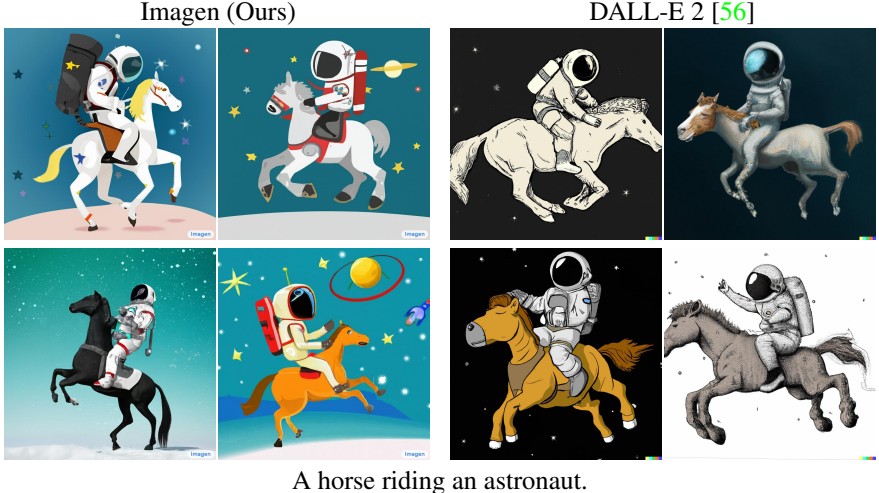

A horse riding an astronaut.

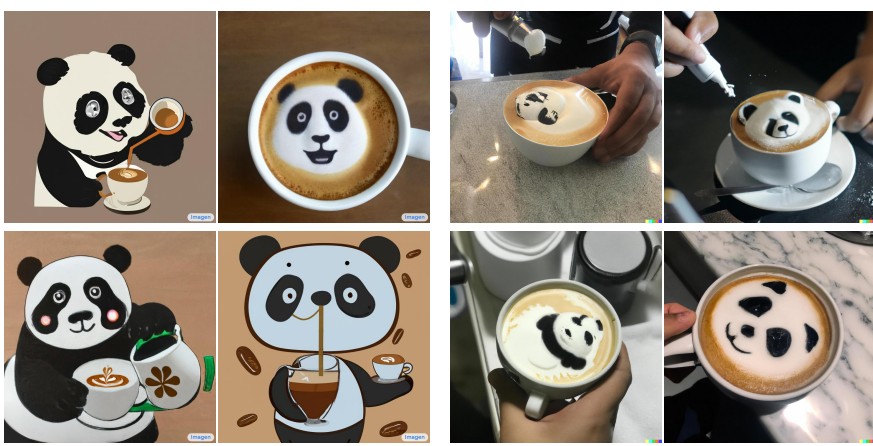

A panda making latte art.

Figure A.19: Example qualitative comparisons between Imagen and DALL-E 2 [56] on DrawBench prompts from Conflicting category. We observe that both DALL-E 2 and Imagen struggle generating well aligned images for this category. However, Imagen often generates some well aligned samples, e.g. "A panda making latte art.".

Imagen (Ours)                    DALL-E 2 [56]

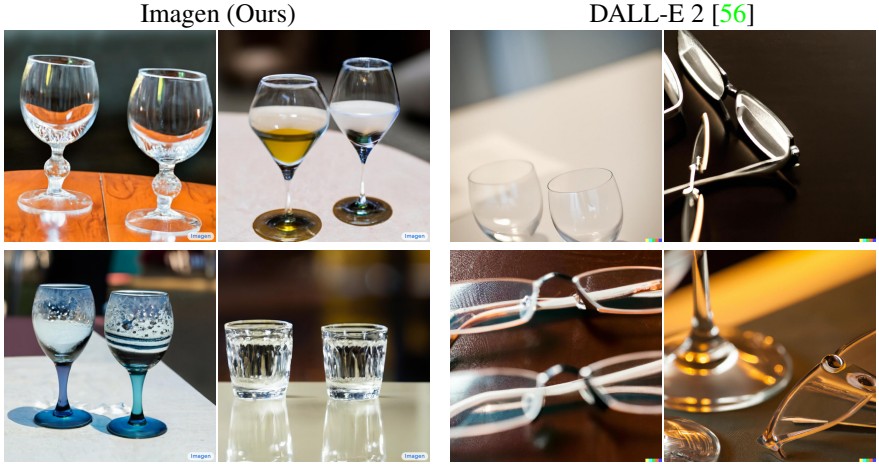

A couple of glasses are sitting on a table.

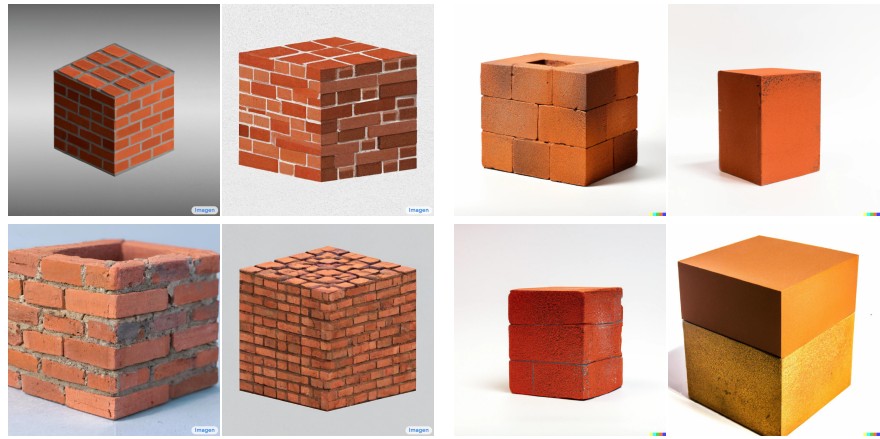

A cube made of brick. A cube with the texture of brick.

Figure A.20: Example qualitative comparisons between Imagen and DALL-E 2 [56] on DrawBench prompts from DALL-E category.

Imagen (Ours)                      DALL-E 2 [56]

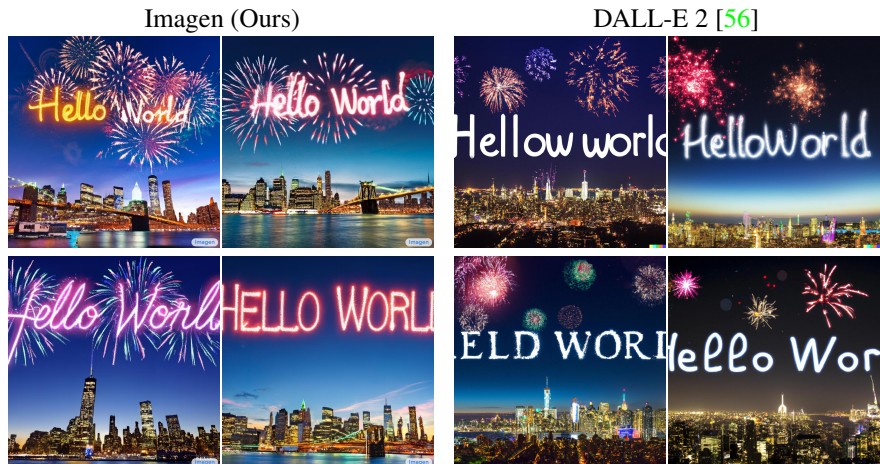

New York Skyline with Hello World written with fireworks on the sky.

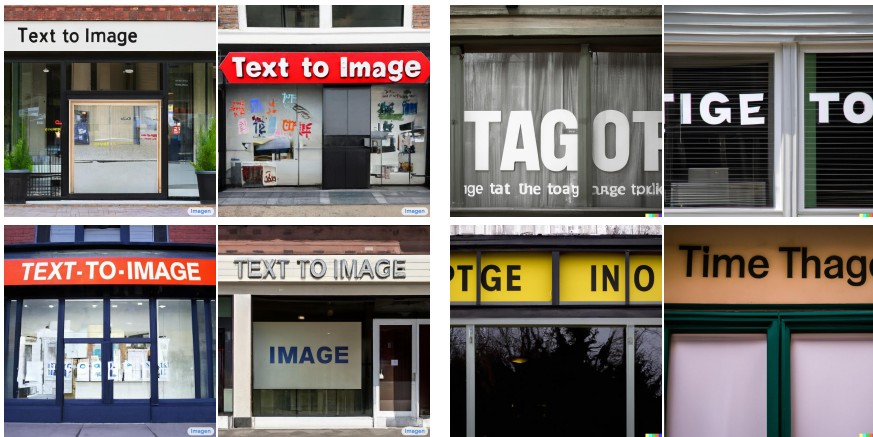

A storefront with Text to Image written on it.

Figure A.21: Example qualitative comparisons between Imagen and DALL-E 2 [56] on DrawBench prompts from Text category. Imagen is significantly better than DALL-E 2 in prompts with quoted text.

Imagen (Ours)                    GLIDE [43]

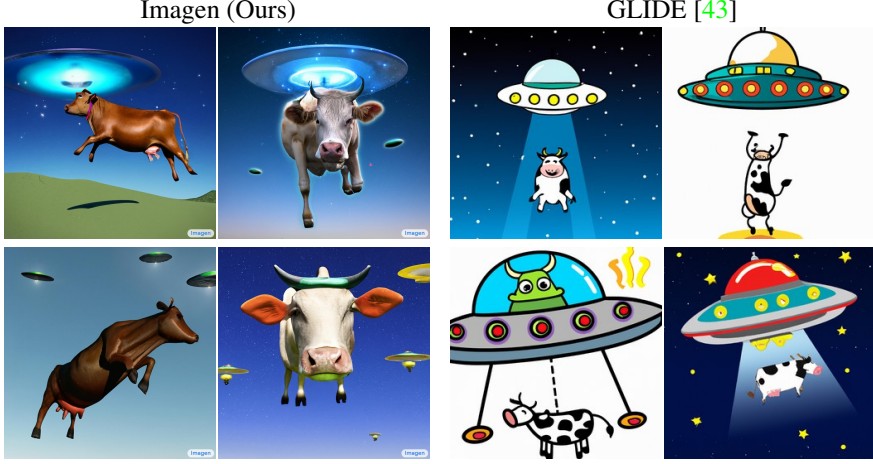

Hovering cow abducting aliens.

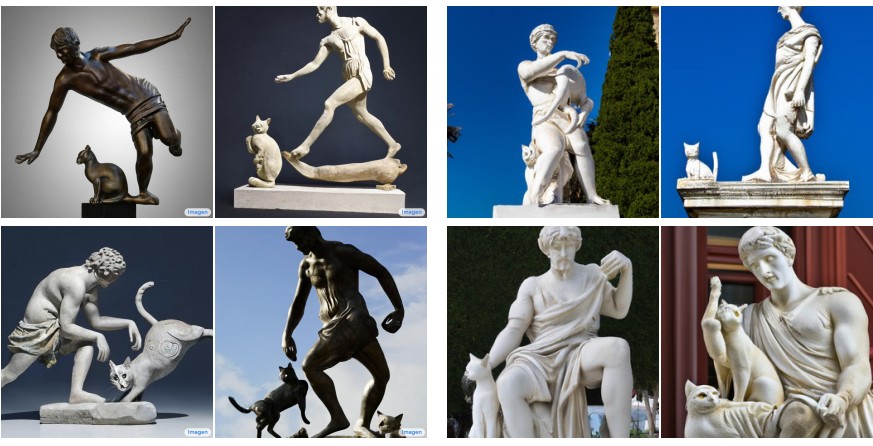

Greek statue of a man tripping over a cat.

Figure A.22: Example qualitative comparisons between Imagen and GLIDE [43] on DrawBench prompts from Reddit category.

Imagen (Ours)              GLIDE [43]

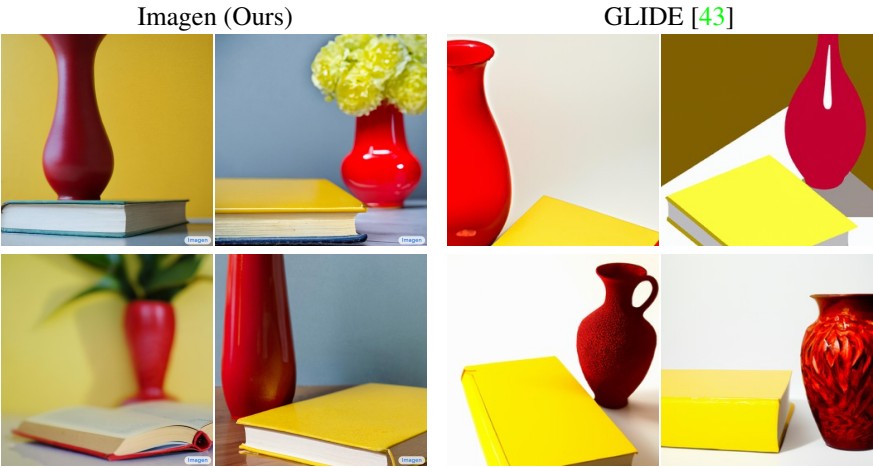

A yellow book and a red vase.

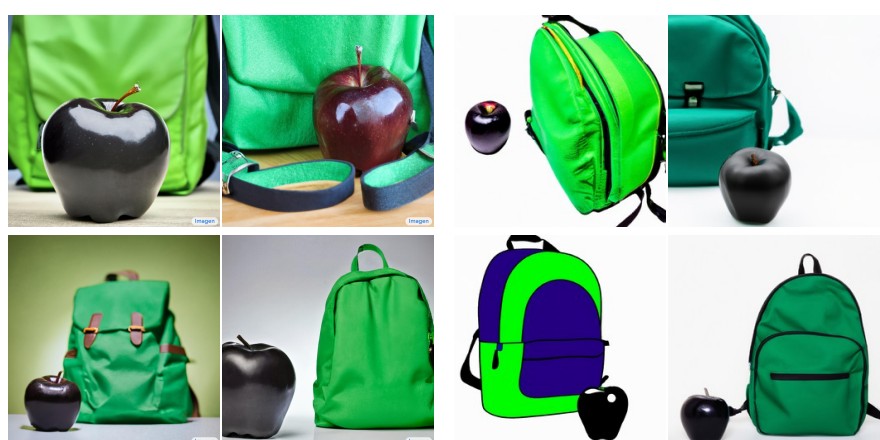

A black apple and a green backpack.

Figure A.23: Example qualitative comparisons between Imagen and GLIDE [43] on DrawBench prompts from Colors category. We observe that GLIDE is better than DALL-E 2 in assigning the colors to the objects.

Imagen (Ours)                          GLIDE [43]

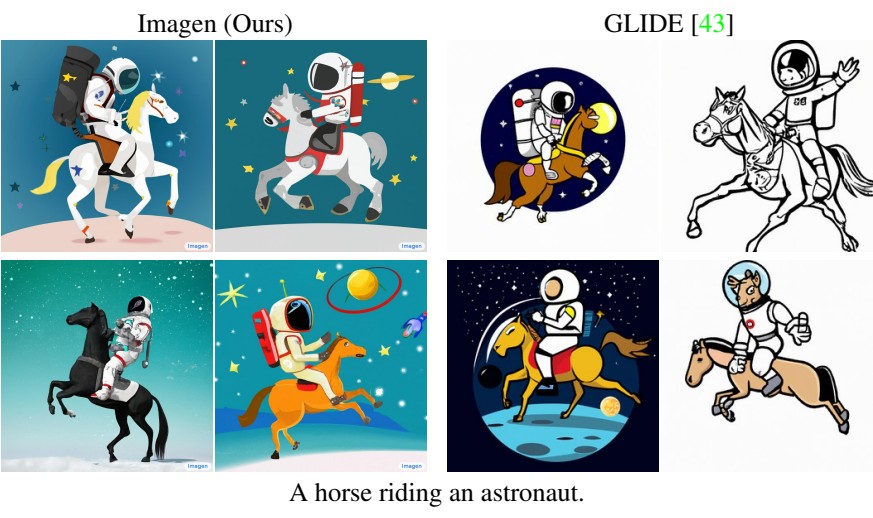

A horse riding an astronaut.

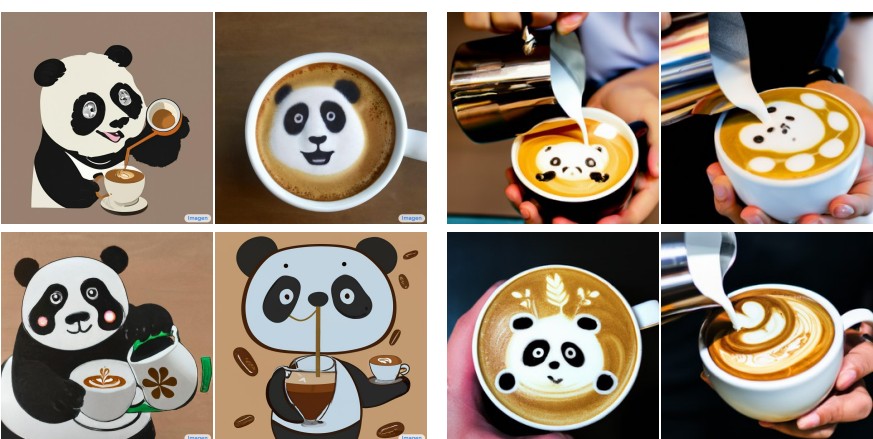

A panda making latte art.

Figure A.24: Example qualitative comparisons between Imagen and GLIDE [43] on DrawBench prompts from Conflicting category.

Imagen (Ours)                              GLIDE [43]

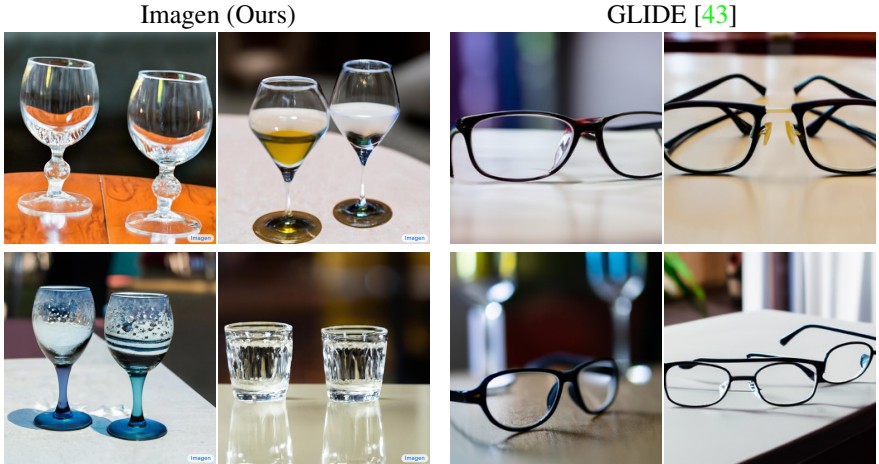

A couple of glasses are sitting on a table.

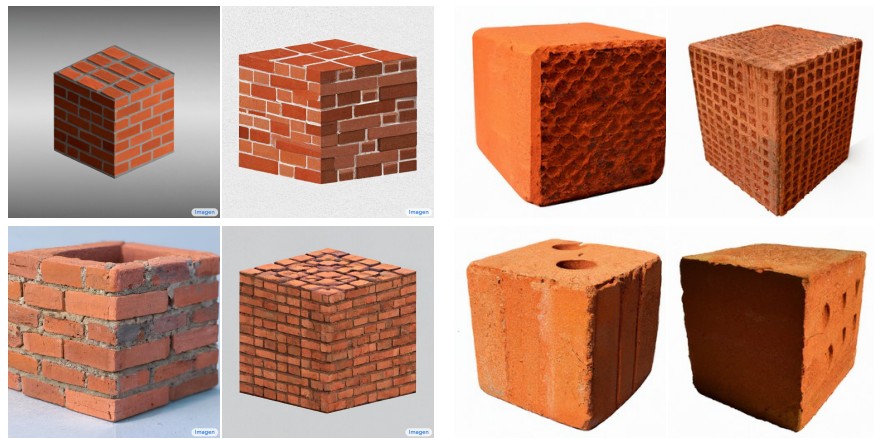

A cube made of brick. A cube with the texture of brick.

Figure A.25: Example qualitative comparisons between Imagen and GLIDE [43] on DrawBench prompts from DALL-E category.

Imagen (Ours)                                    GLIDE [43]

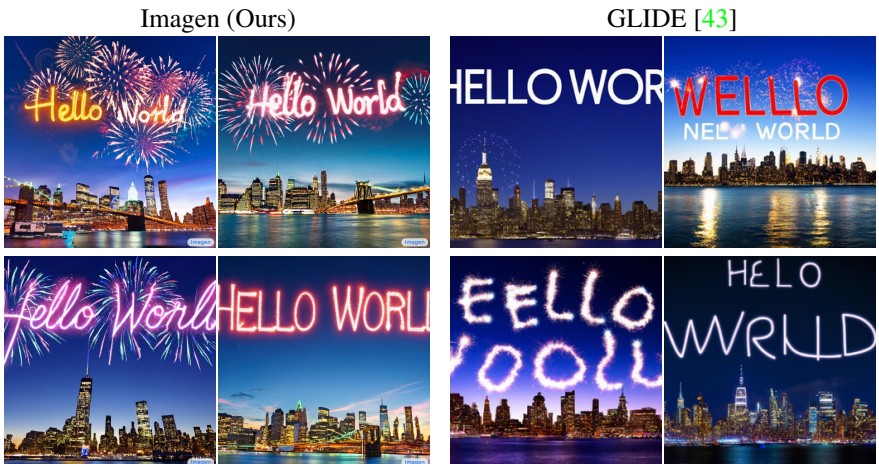

New York Skyline with Hello World written with fireworks on the sky.

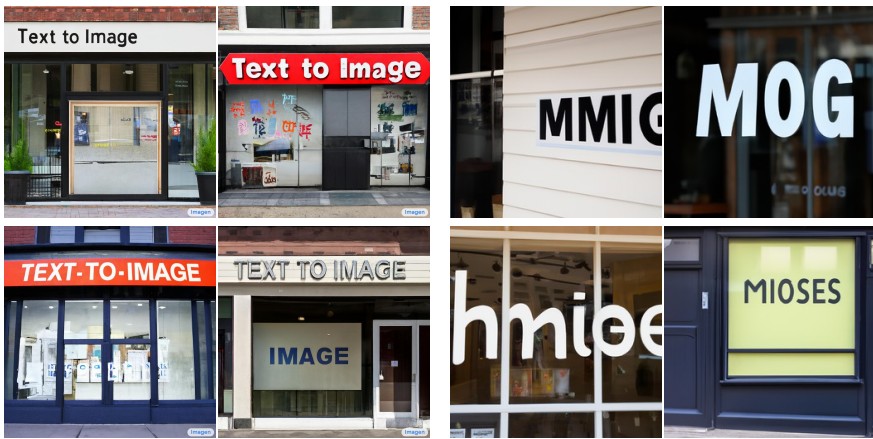

A storefront with Text to Image written on it.

Figure A.26: Example qualitative comparisons between Imagen and GLIDE [43] on DrawBench prompts from Text category. Imagen is significantly better than GLIDE too in prompts with quoted text.

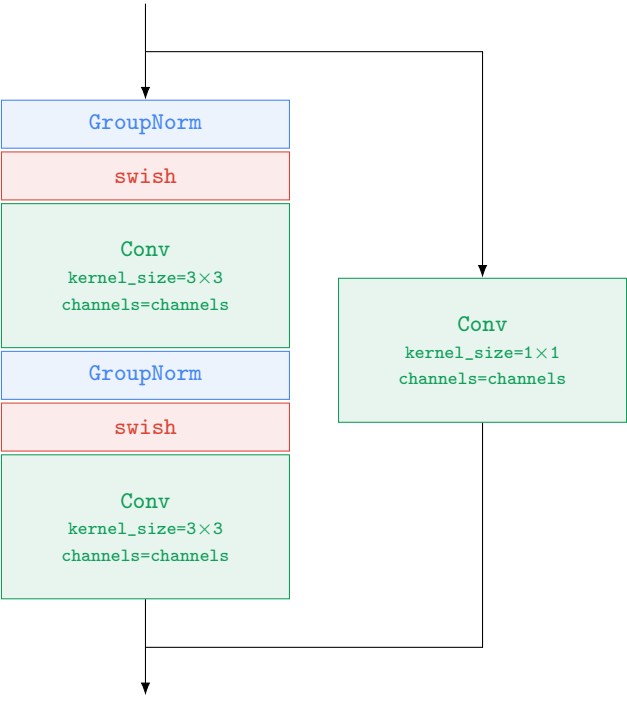

Figure A.27: Efficient U-Net `ResNetBlock`. The `ResNetBlock` is used both by the `DBlock` and `UBlock`. Hyperparameter of the `ResNetBlock` is the number of channels `channels: int`.

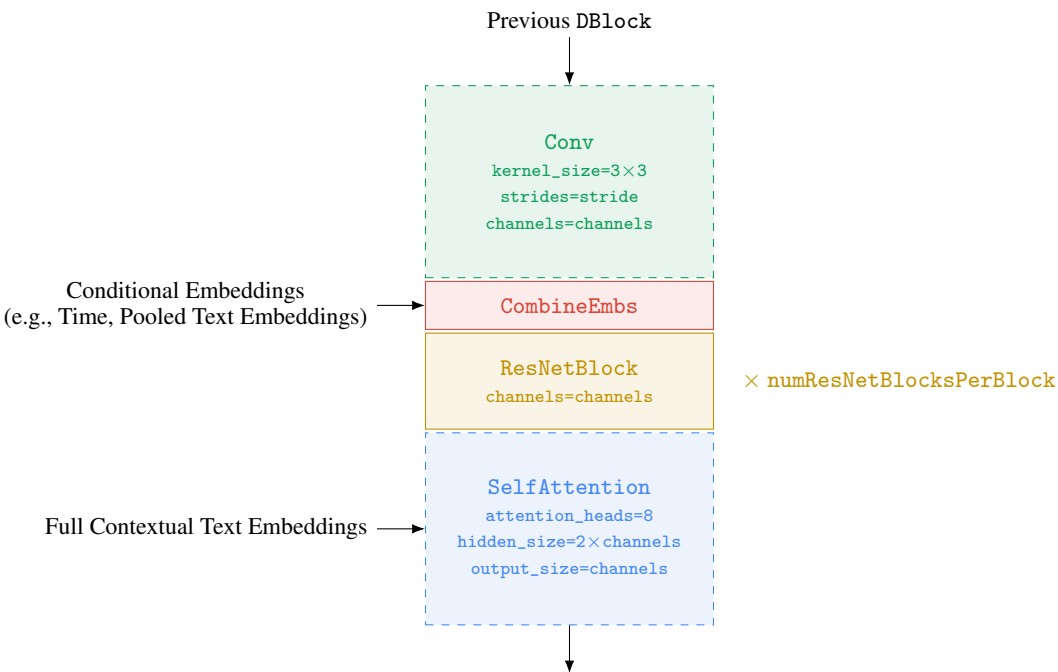

Figure A.28: Efficient UNet `DBlock`. Hyperparameters of `DBlock` are: the stride of the block if there is downsampling `stride: Optional[Tuple[int, int]]`, number of `ResNetBlock` per `DBlock` `numResNetBlocksPerBlock: int`, and number of channels `channels: int`. The dashed lined blocks are optional, e.g., not every `DBlock` needs to downsample or needs self-attention.

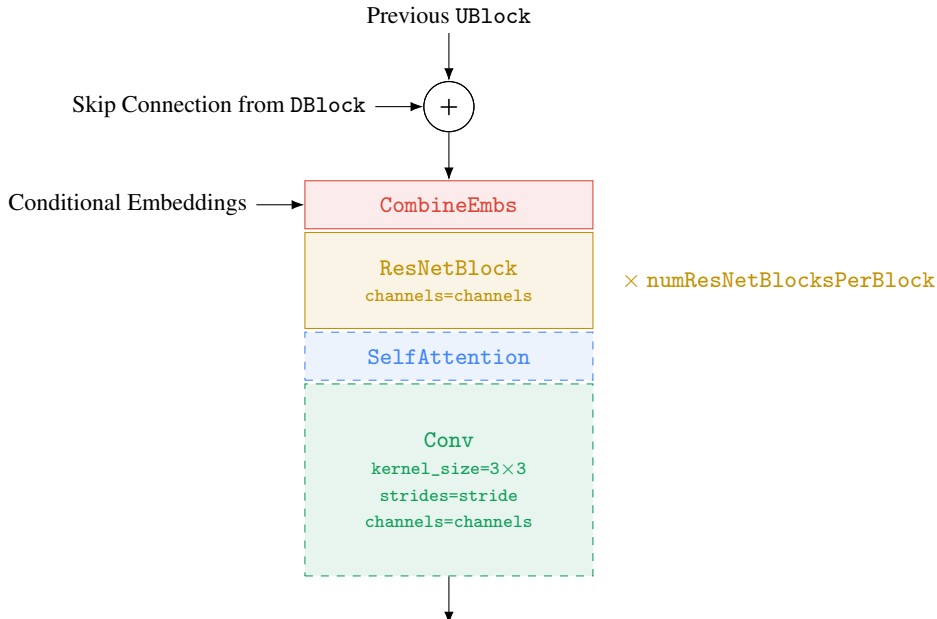

Figure A.29: Efficient U-Net `UBlock`. Hyperparameters of `UBlock` are: the stride of the block if there is upsampling `stride: Optional[Tuple[int, int]]`, number of `ResNetBlock` per `DBlock` `numResNetBlocksPerBlock: int`, and number of channels `channels: int`. The dashed lined blocks are optional, e.g., not every `UBlock` needs to upsample or needs self-attention.

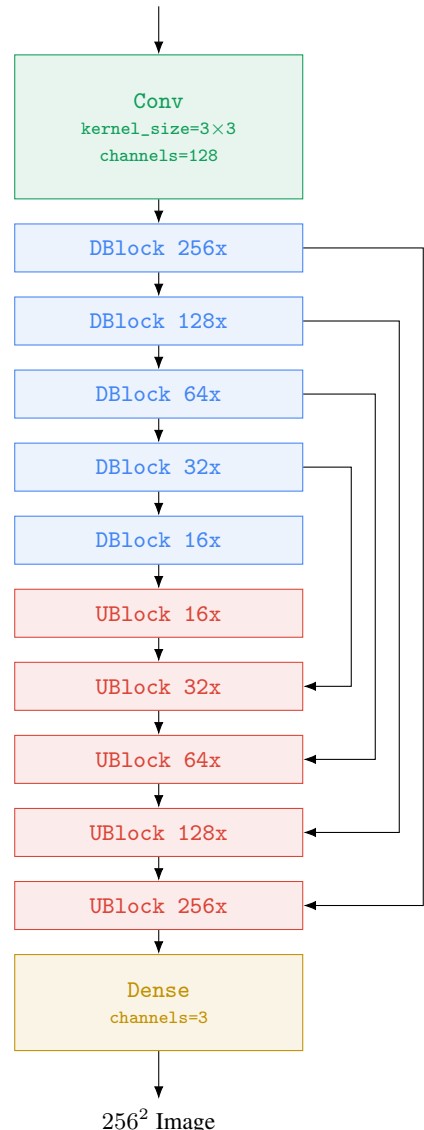

Figure A.30: Efficient U-Net architecture for $64^2 \rightarrow 256^2$.

```
def sample():
  for t in reversed(range(T)):
    # Forward pass to get x0_t from z_t.
    x0_t = nn(z_t, t)

    # Static thresholding.
    x0_t = jnp.clip(x0_t, -1.0, 1.0)

    # Sampler step.
    z_tm1 = sampler_step(x0_t, z_t, t)
    z_t = z_tm1
  return x0_t
```

```
def sample(p: float):
  for t in reversed(range(T)):
    # Forward pass to get x0_t from z_t.
    x0_t = nn(z_t, t)

    # Dynamic thresholding (ours).
    s = jnp.percentile(
        jnp.abs(x0_t), p,
        axis=tuple(range(1, x0_t.ndim)))
    s = jnp.max(s, 1.0)
    x0_t = jnp.clip(x0_t, -s, s) / s

    # Sampler step.
    z_tm1 = sampler_step(x0_t, z_t, t)
    z_t = z_tm1
  return x0_t
```

(a) Implementation for static thresholding.       (b) Implementation for dynamic thresholding.

Figure A.31: Pseudo code implementation comparing static thresholding and dynamic thresholding.

```
def train_step(
    x_lr: jnp.ndarray, x_hr: jnp.ndarray):
  # Add augmentation to the low-resolution image.
  aug_level = jnp.random.uniform(0.0, 1.0)
  x_lr = apply_aug(x_lr, aug_level)

  # Diffusion forward process.
  t = jnp.random.uniform(0.0, 1.0)
  z_t = forward_process(x_hr, t)

  Optimize loss(x_hr, nn(z_t, x_lr, t, aug_level))
```

```
def sample(aug_level: float, x_lr: jnp.ndarray):
  # Add augmentation to the low-resolution image.
  x_lr = apply_aug(x_lr, aug_level)

  for t in reversed(range(T)):
    x_hr_t = nn(z_t, x_lr, t, aug_level)

    # Sampler step.
    z_tm1 = sampler_step(x_hr_t, z_t, t)
    z_t = z_tm1
  return x_hr_t
```

(a) Training using conditioning augmentation.       (b) Sampling using conditioning augmentation.

Figure A.32: Pseudo-code implementation for training and sampling using conditioning augmentation. Text conditioning has not been shown for brevity.

# F   Implementation Details

## F.1   $64 \times 64$

**Architecture**: We adapt the architecture used in [16]. We use larger *embed_dim* for scaling up the architecture size. For conditioning on text, we use text cross attention at resolutions $[32, 16, 8]$ as well as attention pooled text embedding.

**Optimizer**: We use the Adafactor optimizer for training the base model. We use the default optax.adafactor parameters. We use a learning rate of 1e-4 with 10000 linear warmup steps.

**Diffusion**: We use the cosine noise schedule similar to [42]. We train using continuous time steps $t \sim \mathcal{U}(0, 1)$.

```
# 64 X 64 model.
architecture = {
    "attn_resolutions": [32, 16, 8],
    "channel_mult": [1, 2, 3, 4],
    "dropout": 0,
    "embed_dim": 512,
    "num_res_blocks": 3,
    "per_head_channels": 64,
    "res_block_type": "biggan",
    "text_cross_attn_res": [32, 16, 8],
    "feature_pooling_type": "attention",
    "use_scale_shift_norm": True,
}

learning_rate = optax.warmup_cosine_decay_schedule(
    init_value=0.0,
    peak_value=1e-4,
```

```
    warmup_steps=10000,
    decay_steps=2500000,
    end_value=2500000)

optimizer = optax.adafactor(lrs=learning_rate, weight_decay=0)
diffusion_params = {
  "continuous_time": True,
  "schedule": {
    "name": "cosine",
  }
}
```

## F.2  $64 \times 64 \to 256 \times 256$

**Architecture**: Below is the architecture specification for our $64 \times 64 \to 256 \times 256$ super-resolution model. We use an Efficient U-Net architecture for this model.

**Optimizer**: We use the standard Adam optimizer with 1e-4 learning rate, and 10000 warmup steps.

**Diffusion**: We use the same cosine noise schedule as the base $64 \times 64$ model. We train using continuous time steps $t \sim \mathcal{U}(0, 1)$.

```
architecture = {
    "dropout": 0.0,
    "feature_pooling_type": "attention",
    "use_scale_shift_norm": True,
    "blocks": [
        {
          "channels": 128,
          "strides": (2, 2),
          "kernel_size": (3, 3),
          "num_res_blocks": 2,
        },
        {
          "channels": 256,
          "strides": (2, 2),
          "kernel_size": (3, 3),
          "num_res_blocks": 4,
        },
        {
          "channels": 512,
          "strides": (2, 2),
          "kernel_size": (3, 3),
          "num_res_blocks": 8,
        },
        {
          "channels": 1024,
          "strides": (2, 2),
          "kernel_size": (3, 3),
          "num_res_blocks": 8,
          "self_attention": True,
          "text_cross_attention": True,
          "num_attention_heads": 8
        }
    ]
}

learning_rate = optax.warmup_cosine_decay_schedule(
    init_value=0.0,
    peak_value=1e-4,
    warmup_steps=10000,
    decay_steps=2500000,
    end_value=2500000)

optimizer = optax.adam(
    lrs=learning_rate, b1=0.9, b2=0.999, eps=1e-8, weight_decay=0)

diffusion_params = {
  "continuous_time": True,
  "schedule": {
    "name": "cosine",
  }
}
```

**F.3**  $256 \times 256 \rightarrow 1024 \times 1024$

**Architecture**: Below is the architecture specification for our $256 \times 256 \rightarrow 1024 \times 1024$ super-resolution model. We use the same configuration as the $64 \times 64 \rightarrow 256 \times 256$ super-resolution model, except we do not use self-attention layers but rather have cross-attention layers (to the text embeddings).

**Optimizer**: We use the standard Adam optimizer with 1e-4 learning rate, and 10000 linear warmup steps.

**Diffusion**: We use the 1000 step linear noise schedule with start and end set to 1e-4 and 0.02 respectively. We train using continuous time steps $t \sim \mathcal{U}(0, 1)$.

```
"dropout": 0.0,
"feature_pooling_type": "attention",
"use_scale_shift_norm": true,
"blocks"=[
    {
      "channels": 128,
      "strides": (2, 2),
      "kernel_size": (3, 3),
      "num_res_blocks": 2,
    },
    {
      "channels": 256,
      "strides": (2, 2),
      "kernel_size": (3, 3),
      "num_res_blocks": 4,
    },
    {
      "channels": 512,
      "strides": (2, 2),
      "kernel_size": (3, 3),
      "num_res_blocks": 8,
    },
    {
      "channels": 1024,
      "strides": (2, 2),
      "kernel_size": (3, 3),
      "num_res_blocks": 8,
      "text_cross_attention": True,
      "num_attention_heads": 8
    }
]
```