# OpenReview forum: "Photorealistic Text-to-Image Diffusion Models with Deep Language Understanding"
_NeurIPS.cc/2022/Conference — NeurIPS 2022 Accept_

### Official Review · Reviewer_9tx5 · 2022-06-29

**Rating:** 5
**Confidence:** 5
**Ethics Flag:** Yes
**Soundness:** 2 fair
**Presentation:** 3 good
**Contribution:** 2 fair

**Summary:**

This paper uses large transformer language models and diffusion models for text-conditioned image generation. The image generation results are impressive, achieving the state-of-the-art FID results on COCO. This paper uses a dynamic thresholding technique to sample images. This technique achieves better results based on the claims. This paper also builds a new benchmark, DrawBench, for text-to-image tasks.

**Questions:**

The results are impressive, but many text-conditioned image generation works, such as DALLE-2, Parti (just come out), also have good performance. Most of these models use massive data and large models. So it seems like the improvements come from the usage of massive data and large models, which are usually not released.

This paper is a combination of existing techniques, such as language models and diffusion models.
Could the authors explain, except using the massive data and large models, what is the technical contribution of this paper and what is the novelty?

This paper would have much greater impact if it could distinguish itself from other recent/concurrent works. What makes it different from DALLE-2 and how do those differences influence the model's performance and use cases?

Will the authors release their code or data for supporting research in this area?

Is it ok to have such a large figure in the second page. It seems do not fit the NeurIPS template.

**Ethics Review Area:**

["Responsible Research Practice (e.g., IRB, documentation, research ethics)"]

**Limitations:**

This paper claimed their limitations and potential negative societal impact.

(-) This paper has solid results to show the effectiveness of the proposed method. However, the good results are most likely from using large data and large models. This paper simply combines transformer language models and diffusion models, which is not novel. The dynamic thresholding technique is more likely a trick for training models.

(-) This is a good paper with impressive results, but it seems the novelty is limited and not a good match for NeurIPS.


**Strengths And Weaknesses:**

(+) This paper combines transformer language models and diffusion models for image generation. Even though the techniques are not novel, the image generation results are impressive.

(+) This paper observes that using larger transformer language models improves the image generation results significantly.

(+) This paper proposes a new benchmark for evaluating text-to-image tasks.

(-) The novelty is limited. The method is a simple combination of transformer language models and diffusion models.

(-) Using large models and massive data always seems helpful for better results and zero-shot generalization. This has become an obvious "fact" based on recent papers, such as DALLE [1], and CLIP[2]. However, purely having good results without solid methodology contribution is hard to be accepted.

[1] Learning Transferable Visual Models From Natural Language Supervision
[2] Hierarchical Text-Conditional Image Generation with CLIP Latents

---

> ### Author Response · Authors · 2022-08-02
> **Response to the review**
>
> We thank the reviewer for their valuable feedback. We address key comments below.
>
> >The novelty is limited. The method is a simple combination of transformer language models and diffusion models.
>
> We agree that many of the design choices we make are not new. Our work is a novel combination of frozen large language models and cascaded diffusion models, in addition to a novel sampling technique (dynamic thresholding) and novel architecture modifications (efficient U-Net). The novelty of our work is not merely in the design choices themselves, but also in the systematic empirical investigation of various design choices (e.g., the choice of CLIP or T5 text encoder, the impact of size of text encoder and the diffusion models). The magnitude of the improvement we obtain over previous work indicates that our empirical findings are new to the research community.
> We agree that our method is simple, but we consider it as a strength when simple techniques are able to advance the state of the art. We do not require latent variables, image quantization, or similar complexity to achieve high fidelity and strong image-text alignment.
>
> > Using large models and massive data always seems helpful for better results and zero-shot generalization. This has become an obvious "fact" based on recent papers, such as DALLE [1], and CLIP[2] ...
>
> We show detailed ablations that demonstrate the impact of scaling the size of the text encoder over the U-Net model. We believe this is not an obvious fact that scaling the size of the text encoder has more impact than the size of the U-Net model and this is a helpful result for the research community. Furthermore, DALL-E [1] is a 12 billion parameter model, while Imagen is a 3B parameter model that achieves significantly better performance. This shows that scaling is not the only important ingredient for better performance. Fundamental choices such as the choice of generative model, the choice of text encoder, the method of text conditioning etc. have a significant impact on the performance.
>
> > The results are impressive, but many text-conditioned image generation works, such as DALLE-2, Parti (just come out), also have good performance ...
>
> DALL-E 2 is concurrent work. Parti was released on arXiv after the NeurIPS submission deadline. We included significant comparison with DALL-E, but this was not required.
> By comparison, Imagen uses ~ 2x less trainable parameters than DALL-E 2, while achieving better performance on two benchmarks (better MS CoCo FID and better human evaluation on DrawBench). This shows that simply scaling model size may not be the best way to obtain good performance for text to image generation, and technical details are important. Similarly, Parti uses 6x more trainable parameters (20B) to obtain decent text to image synthesis, suggesting that our approach is much more parameter efficient. In addition, upon close inspection of Parti results, one can observe blurry image outputs, which are less desirable than the output of diffusion models such as Imagen and DALL-E 2.
>
> > Could the authors explain, except using the massive data and large models, what is the technical contribution of this paper and what is the novelty?
>
> Our work is a novel combination of frozen large language models and cascaded diffusion models, in addition to a novel sampling technique (dynamic thresholding) and novel architecture modifications (efficient U-Net).
>
> > What makes it different from DALLE-2 and how do those differences influence the model's performance and use cases?
>
> DALL-E 2 is concurrent work and should not be used to minimize our contributions.
>
> Regardless, there are several important differences between Imagen and DALL E-2:
> * Unlike DALL E-2, Imagen does not use a latent prior model. DALL-E requires first training a CLIP model to define the latent space. By contrast, our approach is much simpler and more powerful. We outperform DALLE-2 using approximately half as many trainable parameters.
> * Imagen uses a pretrained frozen language model as a text encoder, while DALL-E 2 uses a CLIP text encoder + another text encoder learned from scratch. We show that using CLIP as a text encoder may harm some capabilities of the model (such as binding attributes to objects, text rendering, etc.). Consequently, Imagen uses a large language model (T5-XXL) as a text encoder which allows deeper language understanding and several compositional capabilities.
> * DALL-E 2 uses relatively small guidance weights, while Imagen introduces dynamic thresholding which allows the capability to use significantly higher guidance weights which enable better text alignment than DALL-E 2 while maintaining a high degree of photorealism.
>
> > Will the authors release their code or data for supporting research in this area?
>
> We refer the reviewer to our Limitations and Societal Impact section regarding our considerations for not releasing the code. A major part of Imagen’s training data is LAION-400M which is public.

---

> > ### Comment · Reviewer_9tx5 · 2022-08-09
> > **Response**
> >
> > The rebuttal addressed some of my concerns, and I like the results shown in this paper. I have raised the score.
> >
> > However, I am not convinced by the author's response that the proposed idea is novel.
> >
> > For example, my question `Could the authors explain, except using the massive data and large models, what is the technical contribution of this paper and what is the novelty?`
> >
> > The response is `Our work is a novel combination of frozen large language models and cascaded diffusion models, in addition to a novel sampling technique (dynamic thresholding) and novel architecture modifications (efficient U-Net).`
> >
> > The combination of the frozen language model and diffusion models is not novel. Many works have used such a combination before, such as GLIDE. The sampling technique is more like an implementation trick.

---

### Official Review · Reviewer_uoqR · 2022-07-08

**Rating:** 8
**Confidence:** 4
**Ethics Flag:** Yes
**Soundness:** 4 excellent
**Presentation:** 4 excellent
**Contribution:** 4 excellent

**Summary:**

Summary:

The paper proposes a text to image diffusion model (Imagen) that generates high fidelity images that accurately match the prompts. The work uses powerful language models (T5-XXL) that have been trained on text corpora that eventually aids in improved language understanding of the final model. Additionally, the paper makes modifications to the existing diffusion models by having dynamic thresholding along with some architectural changes to the U-Net to make it more efficient. With this work, Imagen becomes the state-of-the-art on MS-COCO based on the FID scores surpassing the models that are trained on it. The authors further introduce a benchmark, DrawBench, to evaluate the quality and accuracy of the image synthesis for a battery of challenging prompts.


Strengths:
- The paper makes an important contribution by using the models trained on surplus text corpora (unpaired data) rather than learning from purely paired image-text data. Their experiments also confirm that scaling the pretrained text encoders improves the image generations.
- Imagen model is one of its kind to achieve unprecedented zero-shot results on MSCOCO on the FID metric – surpassing all the existing models that are directly trained on the dataset.
- The paper invents a plethora of ‘tricks’ that facilitated image generations. It includes Efficient U-Net, having dynamic thresholding, text cross-attention layers in the super resolution model, and noise conditioning augmentation. It will be interesting to see if some of these tricks (or principles) can be used for improved generative modeling beyond the scope of the paper.
- The paper introduces DrawBench that aids in evaluating the quality of the model generations across various dimensions. It provides us with a way to systematically compare complementary text to image generative models along these dimensions.
- It was nice to see that the paper uses Human evaluations to compare Imagen with existing models using fidelity and alignment scores. In my experience, CLIPScore might not be a very reliable metric to judge the image-text alignment every time.

Suggestions and Weaknesses:

- [1] showed that scaling the data and the model are important factors in improving performance of zero shot models. The authors show that scaling the model size in terms of text encoder is indeed important for improved performance, however, it might be worthwhile to test how well scaling laws hold in the data dimension. Along the same lines, the Table 1 of the paper compares zero-shot models with different model size and training data. For a fairer comparison of how well the architectural choice is doing, it might be good to compare with an equivalent version of Imagen in terms of model parameters and/or training dataset size.
- Interestingly, the paper shows that Imagen achieves lower preference rate while generating people when compared to the set with no people. It is not clear if the model is finding it difficult to draw people because it has seen a lot of images without people as compared to the images with people in the training dataset or is it ‘inherently’ difficult to create photorealistic people with this architecture? How does Imagen rate when compared to the existing text to image models on the quality of images with and without people?


Typos/Edits:
- Figure 4’s y-axis labels are inconsistent FID-10K vs FID@10K
- Line 338 - wrote ‘Imagen’ two times

Missing references:
- Technically, DALLE-mini also leverages the power of a BART encoder pretrained on text-only corpora. The authors should include it in their related work and discuss it a bit.

*References*

[1] Combined Scaling for Open-Vocabulary Image Classification: https://arxiv.org/pdf/2111.10050.pdf

[2] DALLE-mini: https://wandb.ai/dalle-mini/dalle-mini/reports/DALL-E-Mini-Explained--Vmlldzo4NjIxODA#our-dall-e-model-architecture


**Questions:**

Mentioned in the main review

**Ethics Review Area:**

["Discrimination / Bias / Fairness Concerns"]

**Limitations:**

Mentioned in the main review

**Strengths And Weaknesses:**

Mentioned in the main review

---

> ### Author Response · Authors · 2022-08-02
> **Response to the review**
>
> We thank the reviewer for their valuable feedback. We address key comments below.
>
> > [1] showed that scaling the data and the model are important factors in improving performance of zero shot models. The authors show that scaling the model size in terms of text encoder is indeed important for improved performance, however, it might be worthwhile to test how well scaling laws hold in the data dimension. Along the same lines, the Table 1 of the paper compares zero-shot models with different model size and training data. For a fairer comparison of how well the architectural choice is doing, it might be good to compare with an equivalent version of Imagen in terms of model parameters and/or training dataset size.
>
> We agree with the reviewer that analyzing scaling laws w.r.t the training dataset size is an interesting avenue to explore. We leave this to future work.
>
> > Interestingly, the paper shows that Imagen achieves lower preference rate while generating people when compared to the set with no people. It is not clear if the model is finding it difficult to draw people because it has seen a lot of images without people as compared to the images with people in the training dataset or is it ‘inherently’ difficult to create photorealistic people with this architecture? How does Imagen rate when compared to the existing text to image models on the quality of images with and without people?
>
> Prior work has shown [SR3], diffusion models to be capable of generating photorealistic human faces. However, we believe the dip in performance for images with people comes from 1) limited training data with people, 2) the complex structure of a human face and body (such as hands), and our ability to quickly spot the imperfections. We believe this issue can be resolved by re-weighting the training data to over-represent faces.
>
> It will be difficult to compare Imagen to other text → image models on human generation due to limitations (e.g., DALL-E 2 API ToS limits human generation).
>
> [SR3] Image Super-Resolution via Iterative Refinement, 2022.

---

> > ### Comment · Reviewer_uoqR · 2022-08-04
> > **Response to Author Rebuttal**
> >
> > I thank the authors for their responses. I am fairly content with them.
> > It would be good if they include the missing references in their final version.

---

### Official Review · Reviewer_EZNR · 2022-07-12

**Rating:** 7
**Confidence:** 4
**Soundness:** 4 excellent
**Presentation:** 4 excellent
**Contribution:** 3 good

**Summary:**

This paper proposes Image which is a text-to-image diffusion model. The major finding of this work is that using large language models pretrained only on text data as text encoders are effective for text-to-image generation and can benefit from the scaling power of language models. Dynamic thresholding and Efficient U-Net architecture are proposed to improve the training effectiveness and efficiency of the diffusion model. SOTA experimental results are achieved on COCO FID and the proposed DrawBench benchmark.

**Questions:**

Please refer to the weaknesses for my questions.

**Limitations:**

The limitations are well-addressed.

**Strengths And Weaknesses:**

Strengths:

* The paper is well written and presented.
* The major finding of this work is that using large language models pretrained only on text instead of trained on paired image-text data only on text data is insightful.
* The experiment results in generating photorealistic with text alignment are amazingly impressive.

Weaknesses:
* The technique contributions of this paper are limited. The proposed dynamic thresholding heuristic and U-Net architecture are somehow technically incremental. However, dynamic thresholding seems to be very effective for training the diffusion model. If some theoretical justification can be included, I will increase my rating.

* It is not clear if we should pursue the text language encoding models or paired image-text models if we have enough image and text data. In another word, if we have enough paired image-text data in the future, can the paired image-text encoding models outperform the text language encoding models.

---

> ### Author Response · Authors · 2022-08-02
> **Response to the review**
>
> We thank the reviewer for their valuable feedback. We address key comments below.
>
> > The technique contributions of this paper are limited. The proposed dynamic thresholding heuristic and U-Net architecture are somehow technically incremental. However, dynamic thresholding seems to be very effective for training the diffusion model. If some theoretical justification can be included, I will increase my rating.
>
> We emphasize that improvements such as dynamic thresholding and U-Net architecture are key critical components to making Imagen work. Furthermore, such general techniques for improving diffusion samplers and making the neural net architectures efficient can be helpful for many other conditional and unconditional diffusion models.
> We also emphasize other technical contributions, such as a detailed analysis of different types of text encoders for text to image generation, scaling laws for text encoders and U-Net architectures, and a new benchmark for evaluating text to image models.
>
> > It is not clear if we should pursue the text language encoding models or paired image-text models if we have enough image and text data. In another word, if we have enough paired image-text data in the future, can the paired image-text encoding models outperform the text language encoding models.
>
> It is difficult to procure paired “image and text data” and we will always have more text-only data compared to paired image-text data. Additionally, even if there is comparable data, it is difficult to scale the size of the text encoder on an image-text model (e.g., CLIP) -- this is due to the memory consumption of the image tower.

---

> > ### Comment · Reviewer_EZNR · 2022-08-05
> > **CLIP v.s. Imagen**
> >
> > Thanks for the author's response. It is indeed that text-only data is always more accessible than paired image-text data. It is clear that Imagen benefits from utilizing large-scale text-only data. But I suggest the authors clearly emphasize the technical findings of this paper and avoid guiding the community to incorrect conclusions. To be more specific, if we restrict we have the same amount of text data as CLIP which paired image data is available. Should we train the text-to-image generative models in the CLIP fashion or Imagen fashion?

---

> > > ### Author Response · Authors · 2022-08-08
> > > **Author Response**
> > >
> > > We apologize if the message was not clearly stated in the paper. Through the comparison between text only language models and image-text models as text encoders, we wanted to emphasize that at the current state of the community, where language models are much bigger than image-text models and are trained on much larger datasets, they may be a more promising class of models to be used as an off-the-shelf text encoder.
> > > We agree with the reviewer that a comparison between text-only vs image-text encoders over various amounts of data would be interesting. The ideal chart would be tradeoff FID-CLIP curves of various encoders trained on different amounts of data and seeing their results. We leave this to future work.

---

### Review · Ethics_Reviewer_bx7q · 2022-08-05

**Recommendation:**

The authors might describe what criteria need to be met if a model similar to this one would be released publicly in the future. The authors might also want to consult https://openai.com/blog/best-practices-for-deploying-language-models/ if they do seek to release the model at some point.

**Ethical Issues:**

Yes

**Ethics Review:**

The paper describes a text-to-image diffusion model and benchmark for assessment of such models. These models rely on LLMs for image generation, often a significant source of ethics issues already well documented by other researchers. The authors have chosen to not release this model publicly due to the potential issues for unintentional harm, abuse and intentional misuse.

---

### Meta-Review · Area_Chair_fxKq · 2022-08-20

**Recommendation:** Accept
**Confidence:** Certain

**Metareview:**

This paper proposes Imagen that uses large transformer language models and diffusion models for text-to-image generation. The major finding is that using large language models pretrained only on text data as text encoders are effective. Dynamic thresholding and Efficient U-Net architecture are proposed to improve the training effectiveness and efficiency of the diffusion model.

It received scores of 578. All the reviewers agree that the image generation results are impressive, and the zero-shot results on COCO are strong. This paper also proposes a new benchmark for comprehensively evaluating text-to-image tasks. On the other hand, Reviewer 9tx5 pointed out that one major concern is that the novelty is quite limited.

Overall, the AC thinks that the paper presented impressive results and has great significance, therefore, the AC would like to recommend acceptance of the paper.

**Award:**

Yes

---

### Decision · Program_Chairs · 2022-09-14

Accept